# Generalized and scalable trajectory inference in single-cell omics data with VIA

Shobana V. Stassen [1], Gwinky G. K. Yip [1], Kenneth K. Y. Wong [1,2], Joshua W. K. Ho [3,4] & Kevin K. Tsia [1,2✉]

Inferring cellular trajectories using a variety of omic data is a critical task in single-cell data science. However, accurate prediction of cell fates, and thereby biologically meaningful discovery, is challenged by the sheer size of single-cell data, the diversity of omic data types, and the complexity of their topologies. We present VIA, a scalable trajectory inference algorithm that overcomes these limitations by using lazy-teleporting random walks to accurately reconstruct complex cellular trajectories beyond tree-like pathways (e.g., cyclic or disconnected structures). We show that VIA robustly and efficiently unravels the fine-grained sub-trajectories in a 1.3-million-cell transcriptomic mouse atlas without losing the global connectivity at such a high cell count. We further apply VIA to discovering elusive lineages and less populous cell fates missed by other methods across a variety of data types, including single-cell proteomic, epigenomic, multi-omics datasets, and a new in-house single-cell morphological dataset.

[1] Department of Electrical & Electronic Engineering, The University of Hong Kong, Pokfulam, Hong Kong. [2] Advanced Biomedical Instrumentation Centre, Hong Kong Science Park, Shatin, New Territories, Hong Kong. [3] School of Biomedical Sciences, Li Ka Shing Faculty of Medicine, The University of Hong Kong, Pokfulam, Hong Kong. [4] Laboratory of Data Discovery for Health, Hong Kong Science Park, Shatin, New Territories, Hong Kong. ✉email: tsia@hku.hk

Single-cell omics data captures snapshots of cells that catalog cell types and molecular states with high precision. These high-content readouts can be harnessed to model evolving cellular heterogeneity and track dynamical changes of cell fates in tissue, tumor, and cell population. However, current computational methods face four critical challenges. First, it remains difficult to accurately reconstruct high-resolution cell trajectories and automatically detect the pertinent cell fates and lineages without relying on prior knowledge of input parameter settings. This is a foundational but unmet attribute of trajectory inference (TI) that could make lineage prediction less biased towards input parameters, and thus minimize the confounding factors that impact the underlying hypothesis testing. However, even the few algorithms which automate cell fate detection (e.g., SlingShot[1], Palantir[2], STREAM[3], and Monocle3[4]) exhibit low sensitivity to cell fates and are highly susceptible to changes in input parameters. Second, current trajectory inference (TI) methods predominantly work well on tree-like trajectories (e.g., Slingshot and STREAM), but lack the generalizability to infer disconnected, cyclic or hybrid topologies without imposing restrictions on transitions and causality[5]. This attribute is crucial in enabling unbiased discovery of complex trajectories which are commonly not well known a priori, especially given the increasing diversity of single-cell omic datasets. Third, the growing scale of single-cell data, notably cell atlases of whole organisms[4,6], embryos[7,8], and human organs[9], exceeds the existing TI capacity, not just in runtime and memory, but in preserving both the fine-grain resolution of the embedded trajectories and the global connectivity among them. Very often, such global information is lost in current TI methods after extensive and multiple rounds of dimension reduction or subsampling. Fourth, fueling the advance in single-cell technologies is the ongoing pursuit to understand cellular heterogeneity from a broader perspective beyond transcriptomics. A notable example is the emergence of single-cell imaging technologies that now allow information-rich profiling of morphological and biophysical phenotypes of single cells, and thus offer mechanistic cues to cellular functions that cannot be solely inferred by proteomic or sequencing data (e.g., in cancer[10], ageing[11], and drug responses[12]). However, the applicability of TI to a broader spectrum of single-cell data has yet to be fully exploited.

To overcome these recurring challenges, we present VIA, a graph-based TI algorithm that uses a new strategy to compute pseudotime, and reconstruct cell lineages based on lazy-teleporting random walks integrated with Markov chain Monte Carlo (MCMC) refinement (Fig. 1). VIA relaxes common constraints on traversing the graph, and thus allows capture of cellular trajectories not only in multi-furcations and trees, but also in disconnected and cyclic topologies. The lazy-teleporting MCMC characteristics also make VIA robust to a wide range of pre-processing and input algorithmic parameters, and allow VIA to consistently identify pertinent lineages that remain elusive or even lost in other top-performing and popular TI algorithms we benchmark[5], which are chosen for comparative analysis conditional on meeting several of the following criteria: automated lineage path and cell fate prediction, recovery of complex topologies not limited to trees, scalability and generalizability to multiple single-cell-modalities. We validate the performance of VIA and thus its ability to offer better interpretation of the underlying biology across a variety of transcriptomic, epigenomic, and integrated multi-omic datasets (seven biological datasets with a further two datasets presented in Supplementary). Notably, we show in subsequent sections that VIA accurately detects minor dendritic sub-populations and their characteristic gene expression trends in human hematopoiesis; automatically identifies pancreatic islets including rare delta cells; and recovers endothelial

and cardiomyocyte bifurcation in integrated data sets of single-cell RNA-sequencing (scRNA-seq) and single-cell sequencing assay for transposase-accessible chromatin (scATAC-seq).

Another defining attribute of VIA is its resilience in handling the wide disparity in single-cell data size, structure and dimensionality across modalities. Specifically, VIA is highly scalable with respect to number of cells ($10^2$ to >$10^6$ cells) and features, without requiring extensive dimensionality reduction or subsampling which compromise global information. Most TI methods require two stages of dimensionality reduction in the form of PCA followed by a subsequent stage of UMAP, MLLE, or diffusion components. Only a low number of components from the second layer of dimensionality reduction is retained as an input to the TI method (e.g., STREAM, Monocle3, Slingshot, and even PAGA and Palantir which subset the diffusion components after PCA). In VIA, we show that for cytometry data there is no need for any dimensionality reduction, and for transcriptomic data we show that VIA does not need a second dimensionality reduction step but robustly infers lineages on a wide range of input principal components (PCs). Although PCA is a common step in analyzing transcriptomic data in order to strengthen the signal in the data, we also show that in-principle, VIA can handle 1000 s of genes as direct inputs without any PCA at all (Supplementary Note 5 and Figs. 27–29). We showcase the scalability of sample size by analyzing the fine-grained developmental sub-trajectories in the 1.3-million-cell mouse organogenesis atlas in terms of fast runtime and preservation of global cell-type connectivity, which is otherwise lost in existing TI methods. We also show that VIA is robust against the dimensionality drop (down to 10's–100's antibodies or morphological features) in mass cytometry (proteomics) and imaging cytometry (morphological) data. For instance, VIA consistently reconstructs the pseudotime that recapitulates murine embryonic stem cells (ESCs) differentiation toward mesoderm cells in CyTOF data, where the lazy-teleporting MCMCs contribute to the high accuracy of inference. Lastly, we hypothesize that VIA can also be applied to imaging cytometry for gaining a mechanistic biophysical understanding of cellular progress. To this end, we profiled the biophysical and morphological phenotypes of single-cell live breast cancer cells with our recently developed high-throughput imaging flow cytometer, called FACED[13]. Validated with the in situ fluorescence (FL) image capture, we found that VIA reliably reconstructs the continuous cell-cycle progressions from G1-S-G2/M phase, and reveals subtle changes in cell mass accumulation.

## Results

**Algorithm.** VIA first represents the single-cell data as a cluster graph (i.e., each node is a cluster of single cells), computed by our recently developed data-driven community-detection algorithm, PARC, which allows scalable clustering whilst preserving global properties of the topology needed for accurate TI[14] (Step 1 in Fig. 1). The root (starting point) is designated by the user, either as a single-cell index or using group or cluster level labels. The cell fates and their lineage pathways are then computed by a two-stage probabilistic method, which is the key algorithmic contribution of this work (Step 2 in Fig. 1, see "Methods" for detailed explanation). In the first stage of Step 2, VIA models the cellular process as a modified random walk that allows degrees of *laziness* (remaining at a node/state) and *teleportation* (jumping to any other node/state) with pre-defined probabilities. The pseudotime, and thus the graph directionality, can be computed based on the theoretical hitting times of nodes (see the theory and derivation in "Methods" and Supplementary Note 2). The lazy-teleporting behavior prevents the expected hitting time from converging to a local distribution in the graph as otherwise occurs in regular

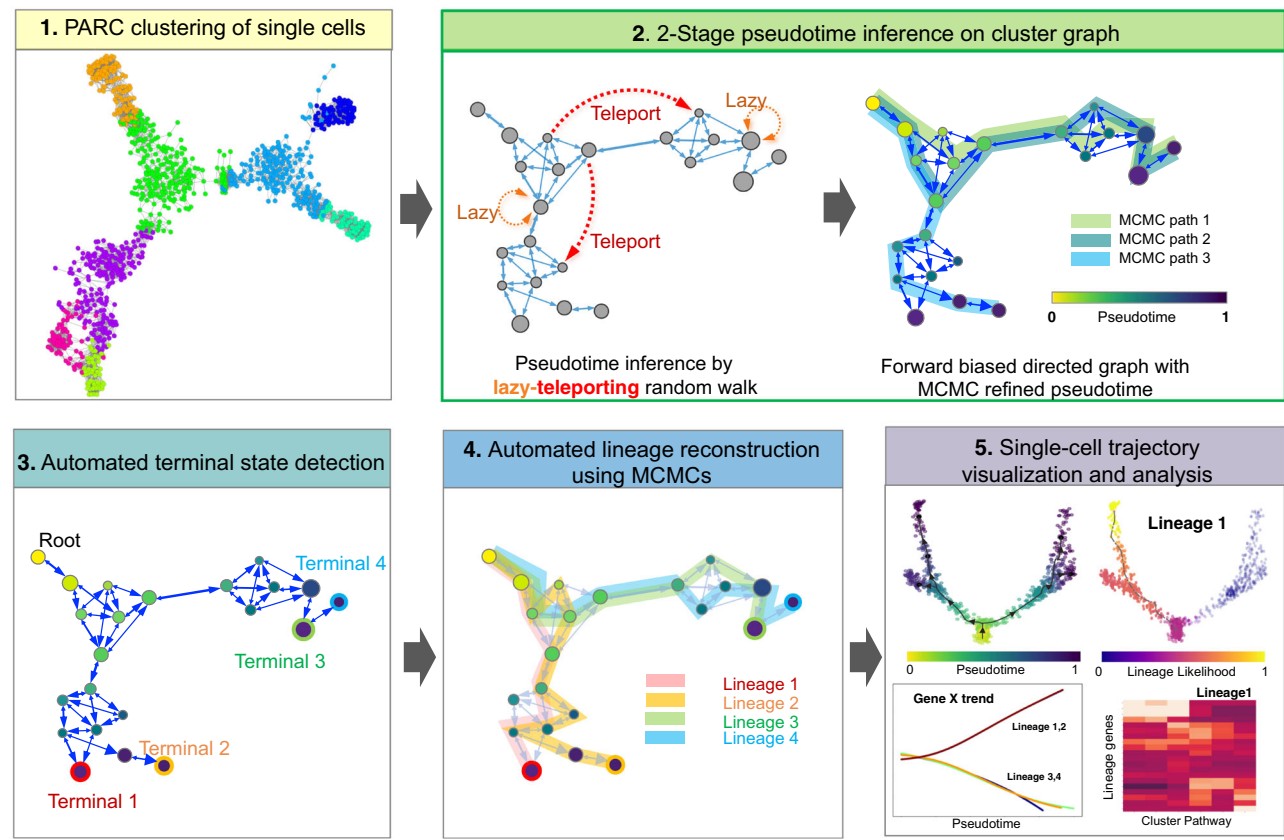

**Fig. 1 General workflow of VIA algorithm.** Step 1: Single-cell level graph is clustered such that each node represents a cluster of single cells (computed by our clustering algorithm PARC[14]). The resulting cluster graph forms the basis for subsequent random walks. Step 2: 2-stage pseudotime computation: (i) The pseudotime (relative to a user defined start cell) is first computed by the expected hitting time for a lazy-teleporting random walk along an undirected graph. At each step, the walk (with small probability) can remain (orange arrows) or teleport (red arrows) to any other state. (ii) Edges are then forward biased based on the expected hitting time (see forward biased edges illustrated as the imbalance of double-arrowhead size). The pseudotime is further refined on the directed graph by running Markov chain Monte Carlo (MCMC) simulations (see three highlighted paths starting at root). Step 3: Consensus vote on terminal states based on vertex connectivity properties of the directed graph. Step 4: lineage likelihoods computed as the visitation frequency under lazy-teleporting MCMC simulations. Step 5: Projection of temporal ordering and lineage probabilities to single-cell level using the original single-cell-KNN graph to enable visualization that combines network topology and single-cell level pseudotime/lineage probability properties onto an embedding using GAMs, as well as unsupervised downstream analysis (e.g., gene expression trend along pseudotime for each lineage).

random walks, especially when the sample size grows[15]. More specifically, the laziness and teleportation factors regulate the weights given to each eigenvector-value pair in the expected hitting time formulation such that the stationary distribution (given by the local-node degree-properties in regular walks) does not overwhelm the global information provided by other "eigen-pairs". Moreover, the computation does not require subsetting the first $k$ eigenvectors (bypassing the need for the user to select a suitable threshold or subset of eigenvectors) since the dimensionality is not on the order of number of cells, but is equal to the number of clusters. Hence all eigenvalue-eigenvector pairs can be incorporated without causing a bottleneck in runtime. Consequently in VIA, the modified walk on a cluster-graph not only enables scalable pseudotime computation for large datasets in terms of runtime, but also preserves information about the global neighborhood relationships within the graph. In the second stage of Step 2, VIA infers the directionality of the graph by biasing the edge-weights with the initial pseudotime computations, and refines the pseudotime through lazy-teleporting MCMC simulations on the forward biased graph.

Next (Step 3 in Fig. 1), the MCMC-refined graph-edges of the lazy-teleporting random walk enable accurate predictions of terminal cell fates through a consensus vote of various vertex connectivity properties derived from the directed graph. The cell

fate predictions obtained using this approach are more accurate and robust to changes in input data and parameters compared to other TI methods (Fig. 2 simulated complex topologies and Supplementary Fig. 1 summary of lineage detection accuracy for all benchmarked real datasets). Trajectories towards identified terminal states are then resolved using lazy-teleporting MCMC simulations (Step 4 in Fig. 1). The single-cell level KNN graph constructed in Step 1 is then used to project the lineage probabilities of trajectories (pathways from root to cell fate), and temporal ordering derived from the cluster-graph topology onto a single-cell level. Together, these four steps facilitate holistic topological visualization of TI on the single-cell level (e.g., using UMAP or PHATE[16,17]) and critically enable data-driven downstream analyses such as recovering gene expression trends and single-cell level pathways of lineages, that are essential to biological validation and discovery of lineage commitment (Methods) (Step 5 in Fig. 1).

**VIA accurately captures complex topologies obscured in other TI methods**. We first generate and analyze simulated datasets (see Methods) to demonstrate that VIA's probabilistic approach to graph-traversal allows it to infer cell fates when the underlying data spans combinations of multifurcating trees and cyclic/disconnected topologies - topologies and lineages often obscured in

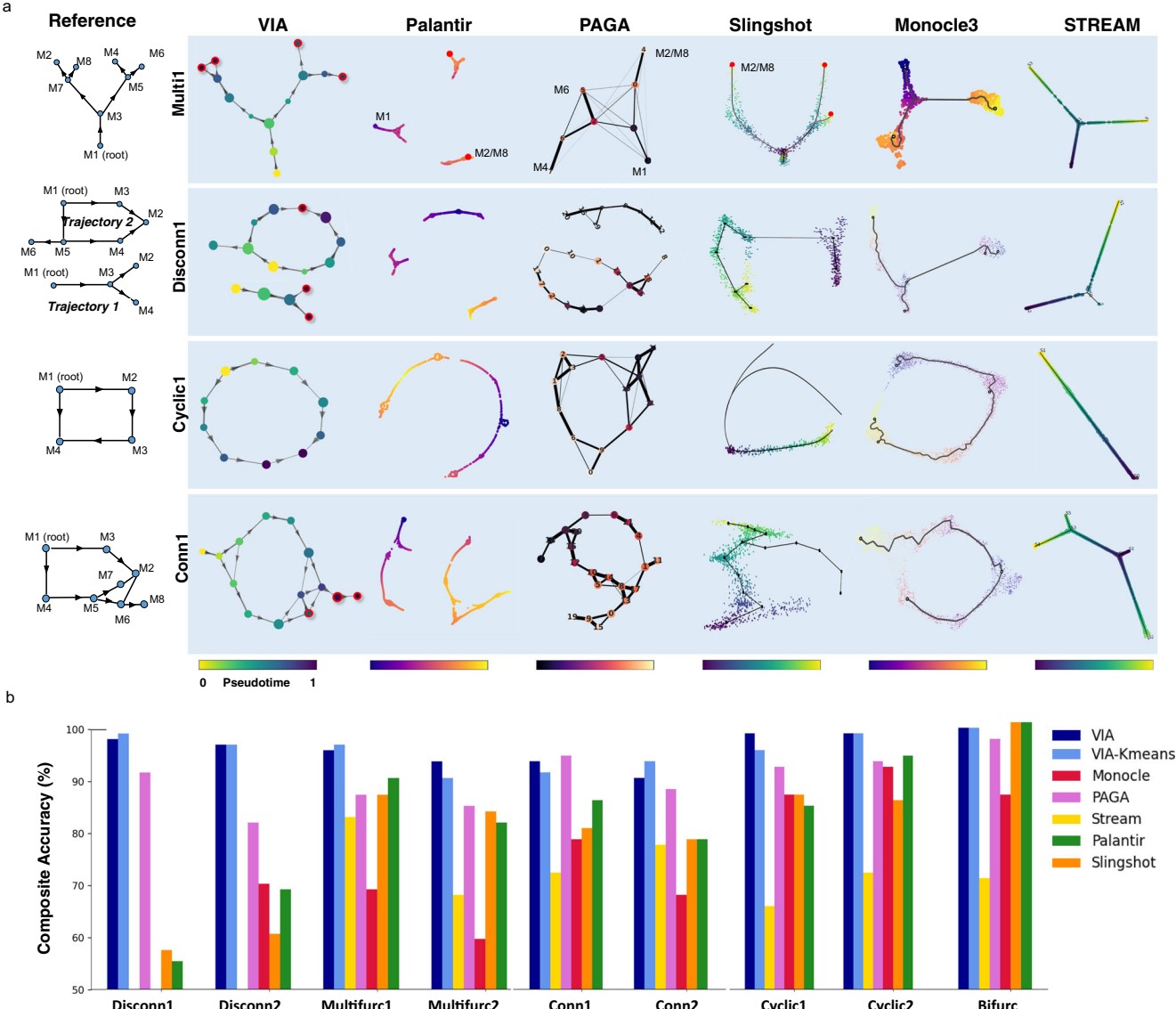

**Fig. 2 TI performance comparisons on complex hybrid topologies. a** Topologies of four representative synthetic datasets (*Multifurc1, Cyclic1, Disconn1,* and *Conn1*) output by different TI methods. The reference topologies are shown on the left. Each dataset contains 1000 "cells" and is run with ten PCs and KNN = 20. VIA is shown at the cluster graph level but can also be projected to the single-cell level as shown in later examples. **b** Composite accuracy score is shown for each method across all nine synthetic datasets (detailed breakdown available in Supplementary Figs. 2–5). Note STREAM does not work on the Disconnected data (producing highly distorted results) and therefore excluded in *Disconn1 and Disconn2*.

existing TI methods. In VIA, the relaxation of edge constraints in computing lineage pathways and pseudotime enables accurate detection of cell fates and complex trajectories by avoiding prematurely imposing constraints on node-to-node mobility. Other methods resort to constraints such as reducing the graph to a tree, imposing unidirectionality by thresholding edges based on pseudotime directionality, removing outgoing edges from terminal states[2,18], and computing shortest paths for pseudotime[1,2].

The availability of a reference truth model for the synthetics datasets allows us to quantify TI accuracy using a composite metric which assesses multiple layers of the inferred trajectory including topology, pseudotime and lineage prediction. The metric assesses "local" graph similarity between the inferred and reference graphs using the Graph Edit Distance (GED) and an F1-Branch score (which labels branches in the inferred topology as true or false positives, or the lack thereof as a false negative). "Global" graph similarity is computed using the Ipsen–Mikhailov metric[19] (Methods), and pseudotime quality is captured by the

Pearson correlation between the inferred and reference pseudotimes. Terminal cell fate prediction is evaluated using the F1-score. The breakdown of the composite score and further detail on each metric is available in Supplementary Note 3 and Supplementary Figs. 2–5.

The differences in accuracy between VIA and other methods is most significant for complex topologies, particularly those with disconnected components comprising various connected topologies, whilst the ability to accurately detect cell fates is highlighted by multilineage furcating topologies. In the four-leaf multifurcation (Fig. 2a top), VIA accurately captures the two cascading bifurcations which lead to four leaf nodes. In particular, VIA detects the elusive "M2" terminal state whereas other methods (Palantir, PAGA, Slingshot, STREAM, and Monocle3) merge it with the "M8" lineage. Monocle3 and STREAM typically only capture a single bifurcation and thus merge the pairs of leaves that otherwise arise from the second layer of bifurcation (Fig. 2a). Even for the fairly simple cyclic topology (Fig. 2a), other methods

tend to fragment the structure to varying degrees depending on the parameter choice whereas VIA consistently preserves the global cyclic structure (Supplementary Fig. 4c under various K (KNN)). This is not to say VIA is invariant to parameter choice, but rather that VIA predictably modulates the graph resolution across a wide range of K without disrupting the underlying global topology (see the increase in the number of nodes in $K = 30$ versus $K = 5$ in Supplementary Fig. 4c). This characteristic is important for robustly analyzing multiple levels of resolution in complex graph topologies, as also shown in our later investigation of the 1.3-million-cell mouse atlas. The performance comparison for the disconnected hybrid topologies (Fig. 2) shows that VIA disentangles the cyclic and bifurcating lineages (that comprise Disconnected1) and captures the key leaf-states in the bifurcation as well as the "tail" extending from the cyclic topology. Palantir overly fragments the two trajectories, whereas Monocle3 and Slingshot merge them, STREAM is not well suited to non-tree trajectories given the underlying structure is assumed to be a spanning tree.

We also show that VIA is flexible to using clustering methods other than PARC by substituting PARC with Kmeans clustering to show that the lazy-teleporting MCMCs still enable faithful recovery of various topologies as well as the associated cell fates (Supplementary Note 6 and Figs. 30–32). The main drawback of using K-means is that under- or over-clustering can occur based on the user-choice of K, whereas methods like PARC enable a more data-driven resolution of the data where the recovery of less populous cell types is not dependent on an adequately large number of clusters.

**VIA reveals rare lineages in epigenomic and transcriptomic landscapes of human hematopoiesis**. To assess the performance of VIA on inferring real cellular trajectory, we first considered a range of scRNA-seq datasets, including hematopoiesis[2,20], endocrine genesis, B-cell differentiation[21], and embryonic stem (ES) cell differentiation in embryoid bodies[17]. We present the analyses of CD34+ human hematopoiesis and endocrine differentiation here, whereas the generalizable performance of VIA on other scRNA-seq datasets is presented in Supplementary Figs. 1, 6, 13. We highlight human hematopoiesis as it has been extensively studied not only with scRNA-seq, but also other single-cell omics modalities, notably scATAC-seq. Hence, it allows us to reliably assess lineage identification performance and downstream analyses using VIA.

First, we show that VIA consistently reveals from the scRNA-seq dataset the typical hierarchical bifurcations during hematopoiesis that result in key committed lineages of hematopoietic stem cells (HSCs) to monocytic, lymphoid, erythroid, classical and plasmacytoid dendritic cell (cDCs and pDCs) lineages and megakaryocytes (Fig. 3a). The automated detection of these terminal states in VIA, as quantified by F1-scores on the annotated cells, remains robust to varying the number of neighbors in the KNN graph, and the number of PCs (Fig. 3c). Specifically, VIA's sustained sensitivity to rarer cell types (e.g., DCs and megakaryocytes) can be attributed to a better underlying graph structure where nodes are well delineated by PARC (as rare cell types are well separated by graph pruning in the clustering stage) and edges governing the random walk pathways are not prematurely removed due to restrictions on causality.

In contrast, the sensitivity of Palantir and Slingshot in detecting rarer lineages drops significantly outside a favorable "sweet spot" of parameters. Slingshot can only recover the major cell populations (monocytes, erythroid, and B cells) and confuses the DC populations with the monocytes and the megakaryocytes with the erythroid cells. Palantir can only identify the DCs and

megakaryocytes for a handful of parameter options, whereas VIA achieves this goal across a wider range of parameters (Fig. 3c). To verify that VIA reliably delineates the megakaryocyte, cDC and pDC lineages, we used VIA to automatically plot the lineage specific trends for selected marker genes. We showed that while both DC lineages exhibit elevated *IRF8*, the *CSF1R* is specific to the cDC, and the *CD123* remains elevated for pDCs whereas it is first up-regulated, then down-regulated in cDCs (Fig. 3b and Supplementary Figs. 7–9). Marker genes known to increase along a specific lineage are correlated against the pseudotime along each lineage as an indicator of correct cell ordering (Fig. 3d). The gene trends inferred by each method are provided in Supplementary Fig. 9 to show a side-by-side comparison of nuances in the quality of plotted expressions, such as the presence of cross-talk between distinct lineages, or distortion of the trends due to unrelated cells assimilated into lineages.

We find that VIA's interpretation of the human scATAC-seq profiles (Fig. 3e) mirrors the continuous landscape of scRNA-seq human hematopoietic data (Fig. 3a). We use two common preprocessing pipelines[20,22] (see Methods), intended to alleviate challenges posed by the sparsity of scATAC-seq data, to show that VIA consistently predicts the expected hierarchy of lineages furcating from hematopoietic progenitors to their descendants. The graph topology of VIA (colored by pseudotime) captures the progression of multipotent progenitors (MPPs) toward the lymphoid-primed MPPs (LMPP) and the common myeloid progenitors (CMPs) which in turn give rise to the CLP and MEP lineages respectively. The known joint contribution of LMPPs and CMPs towards the GMP lineage is also captured by the VIA graph. We verified the lineages identified by VIA by analyzing the changes in the accessibility of TF motifs associated with known regulators of the lineage commitments, e.g., *GATA1* (erythroid), *CEBPD* (myeloid) and IRF8 (DCs) (Fig. 3e, Supplementary Fig. 10c). Again, we note that the detection of these lineages is less straightforward in other methods, which generally face a sharp drop in accuracy of detecting relevant cell fates as the input number of PCs exceeds ~50PCs (e.g., Palantir often misses the CLP and monocyte lineages, **see** Supplementary Fig. 6 for Palantir's outputs across parameters and Fig. 3g for the corresponding prediction accuracy). The quality of the lineage pathways and gene trends is indicated in Fig. 3h by the correlation of lineage cell ordering against marker gene expression. Visual comparisons of the topologies and predicted gene trends of each method are shown in Supplementary Fig. 11. We emphasize that VIA's robustness in handling both the scRNA-seq and scATAC-seq datasets demonstrates its unique ability to achieve stable prediction and thus faithful query of the underlying biology without biasing specific sets of input parameters which nontrivially vary across datasets—as also evident from our series of "stress tests" on VIA's performance and the gene-trend comparisons (Supplementary Fig. 1).

**VIA detects small endocrine Delta lineages and Beta subtypes**. We use a scRNA-seq dataset of E15.5 murine pancreatic cells to again examine whether VIA can automatically detect multiple lineages, in particular less populous ones. This data spans all developmental stages from initial endocrine progenitor-precursor (EP) state (low level of *Ngn3*, or *Ngn3*^low^) to intermediate EP (high level of *Ngn3*, or *Ngn3*^high^) and Fev+ states, to terminal states of hormone-producing alpha, beta, epsilon and delta cells[23] (Fig. 4a).

A key challenge in analyzing this dataset is the automated detection of the small delta-cell population (a mere 3% of the total population), which otherwise requires manual assignment in CellRank and Palantir (see Supplementary Figs. 15, 16 for a comparison of topology and automated gene trend plots along

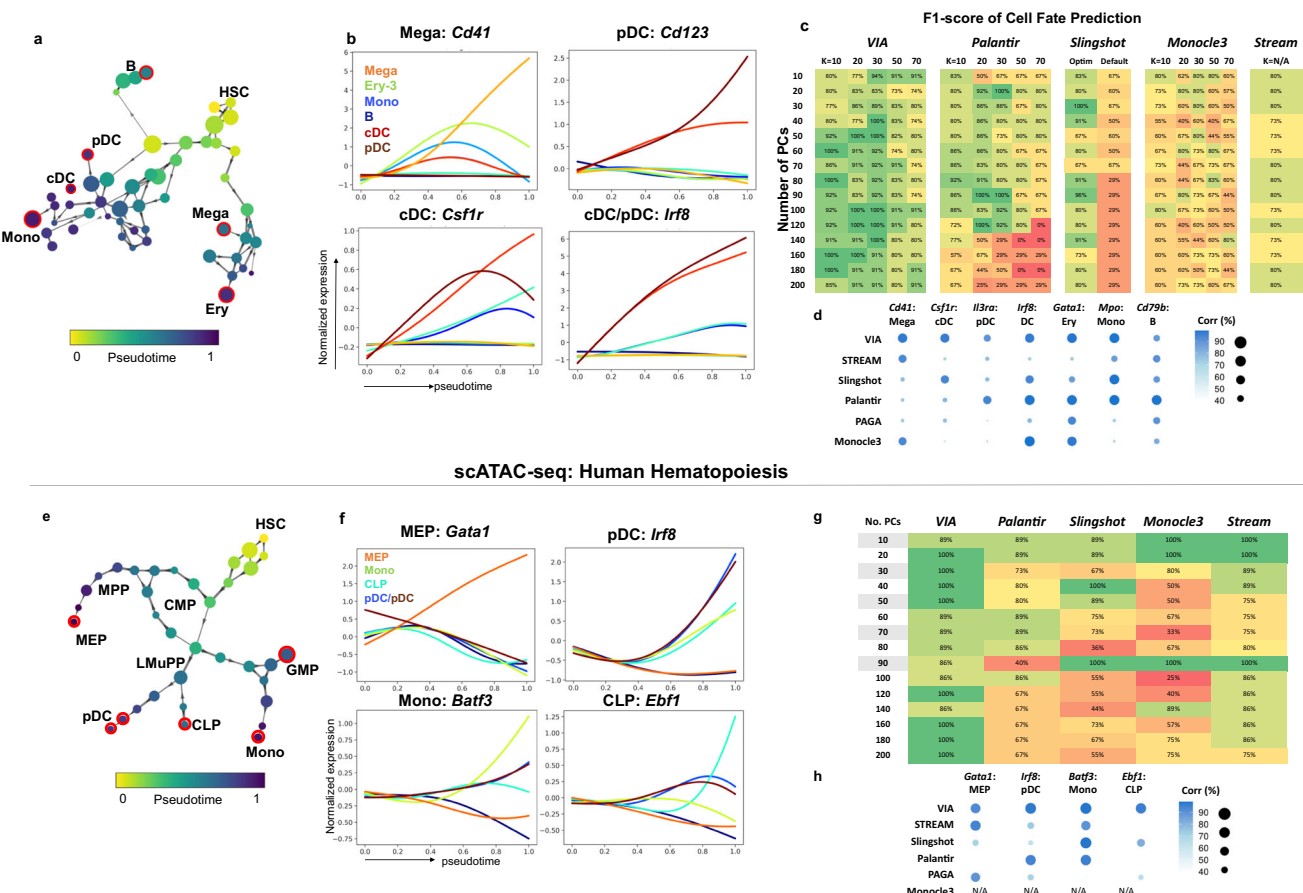

**Fig. 3 VIA analysis of human hematopoiesis based on scRNA-seq and scATAC-seq[20] data. a** VIA graph colored by inferred pseudotime. Identified terminal state nodes are outlined in red and labeled according to their representative annotated cell type (**b**) pseudo-temporal trends of marker genes for key minor populations (see Supplementary Figs. 5–7 for gene trends of all lineages and single-cell pathways) (**c**) F1-scores for terminal state detection of classical and plasmacytoid dendritic cells (mDCs and pDCs), Megakaryocyte (Mega), Erythroid (Ery), Monocyte (Mono), and B-cell lineages (**d**) Pearson correlation of marker gene expression and pseudotime along respective lineages indicate the quality of cell ordering along lineages. A side-by-side comparison of the inferred gene trends by each method provides a more holistic assessment of the quality of expression prediction and can be found in Supplementary Fig. 9 (**e**) Graph topology of scATAC-seq hematopoietic data using Buenrostro[20] preprocessing protocol, nodes colored by inferred pseudotime (**f**) pseudo-temporal trends of transcription-factor motifs (**g**) F1-scores for terminal state detection of megakaryocyte–erythroid progenitor cells (MEP), Common Lymphoid Progenitor (CLP), pDC and Mono lineages for KNN = 20 and different number of PCs. Pre-processed using *k-mer Z Scores* protocol yields a more challenging input as shown by the performance drop for other methods beyond 50PCs. VIA's F1-scores are more robust to choice of number of PCs (**h**) Correlations of gene expression and pseudotime (Full gene-trend and topology comparison in Supplementary Fig. 11).

predicted lineages by different methods). In contrast, the well-delineated nodes of the VIA cluster-graph (a result of the accurate terminal state prediction enabled by the lazy-teleporting MCMC property of VIA on the inferred topology) lends itself to automatically detecting this small population of delta cells, together with all other key lineages (alpha, beta and epsilon lineages) (Fig. 4a–c). As evidenced by the corresponding gene-expression trend analysis, VIA detects all of the hormone-producing cells including delta cells which show exclusively elevated *Hhex, Sst*, and *Cd24a* (Fig. 4c–e). To show that this is not a co-incidence of parameter choice, we verify that these populations can be identified for a wide range of chosen highly variable genes (HVGs prior to PCA) and number of PCs (see Supplementary Fig. 1c). Interestingly, consistent with an observation by Bastidas-Ponce et al.[23] we see two groups of *Fev+* populations branching from the *Ngn+* populations, which subsequently progress towards the distinct cell lines. We show consistency in predicted topology, cell fates and gene trends when applying VIA directly on 1000 s of HVGs without PCA for a wide range of HVGs (see Supplementary Fig. 29), and under artificial degradation of the data to test robustness to noise (see Supplementary

Fig. 18 and Note 4 to see that VIA is more robust to the addition of noise than other methods which merge major lineages).

Interestingly, we find VIA often automatically detects two Beta-cell subpopulations (Beta-1 and Beta-2) (Fig. 4b–e) that express common Beta-cell markers, such as *Dlk1, Pdx1*, but differ in their expressions of *Ins1* and *Ins2* (Fig. 4c–e). The pseudotime order within this Beta-cell heterogeneity[24,25], undetectable by other TI methods (as shown in the gene correlation comparisons Supplementary Fig. 15), can further be reconciled in the VIA graph where the immature Beta-2 population precedes the mature Beta-1 population. We find that the immature Beta-2 population strongly expresses *Ins2*, and weakly expresses *Ins1*, followed by the mature Beta-1 cells which express both types of *Ins*[25] (Fig. 4d–f). VIA graphs colored by *Ins1* and *Ins2* further show the difference in *Ins* expression by the two Beta populations).

**VIA recovers Isl1+ cardiac progenitor bifurcation in multi-omic data.** We next demonstrate the applicability of VIA in single-cell multi-omics analysis by investigating murine *Isl1+* cardiac

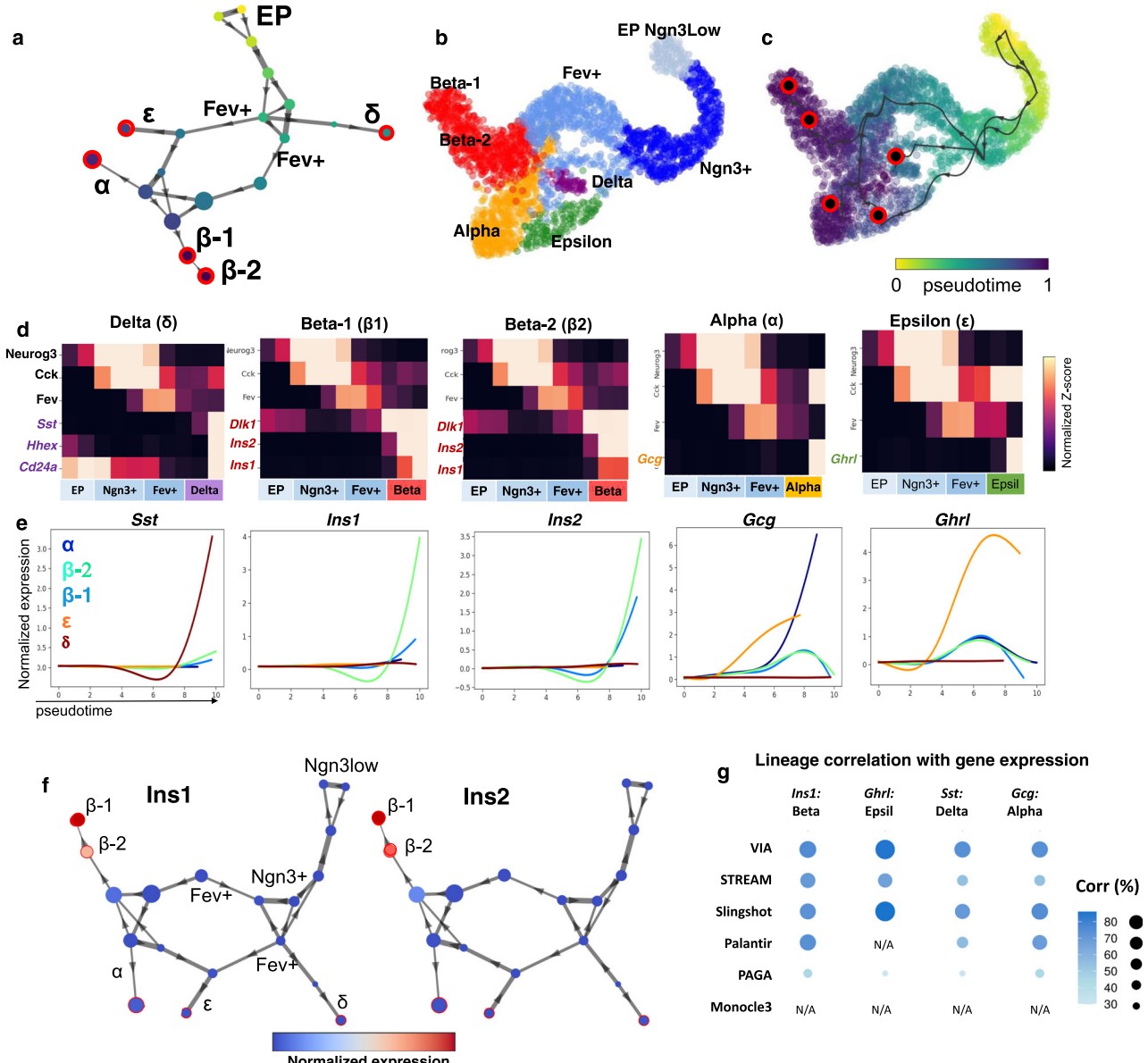

**Fig. 4 VIA detects small populations in endocrine progenitor cell differentiation. a** VIA graph topology Pancreatic Islets: Colored by VIA pseudotime with detected terminal states shown in red and annotated based on known cell type as Alpha, Beta-1, Beta-2, Delta, and Epsilon lineages where Beta-2 is $Ins1^{low}Ins2+$ Beta subtype (Supplementary Fig. 8 for graph node-level gene expression intensity of Ins1 and Ins2). **b** TSNE colored by reference cell type annotations. **c** colored by inferred pseudotime with predicted cell fates in red-black circles. **d** VIA inferred cluster-level pathway shows gene regulation along endocrine progenitor (EP) to $Fev+$ cells followed by expression of islet specific genes. **e** Gene-expression trends along pseudotime for each pancreatic islet. **f** Beta-2 subtype expresses Ins2 but not Ins1, suggestive of an immature Beta cell subtype. **g** Marker gene-pseudotime correlations along respective lineages. Full comparison of gene trends can be referred to Supplementary Fig. 15.

progenitor cells (CPC) which are known to bifurcate towards endothelial and cardiomyocyte fates (Fig. 5). VIA consistently uncovers the bifurcating lineages using both single-cell transcriptomic (scRNA-seq) and chromatin accessibility (scATAC-seq) information[26–28], as well as their data integration (see "Methods" for data integration using Seurat). Other methods that are also applicable to non-transcriptomic data, fail to uncover the two main lineages.

Other methods typically only detect the cardiomyocyte lineage (the inability to detect a bifurcation is exacerbated when the number of input PCs increases), and instead falsely detect several intermediate and early stages as final cell fates. For instance STREAM consistently merges the cardiomyocyte and endothelial lineages and instead presents the intermediate stage as a separate bifurcation. See Supplementary Figs. 20, 21 for sample outputs

across parameters, and Fig. 5g for the corresponding prediction accuracy of each method. PAGA does not offer automated cell fate prediction or lineage paths and is therefore not benchmarked for this dataset. The disparity in trajectory inference is evident in the scRNAseq and integrated data where Monocle3, Slingshot and Palantir do not resolve either of the two cell fates (Fig. 5g), and STREAM detects multiple spurious branches that fragment the structure entirely. We hypothesized that lowering the K (number of nearest neighbors) in Palantir and VIA would be more appropriate given the extremely low cell count (~200 cells) of the scRNA-seq dataset. Whilst this approach did not alter the outcome for Palantir, we found that VIA is able to capture the transition from early to intermediate CPCs and finally lineage committed cells.

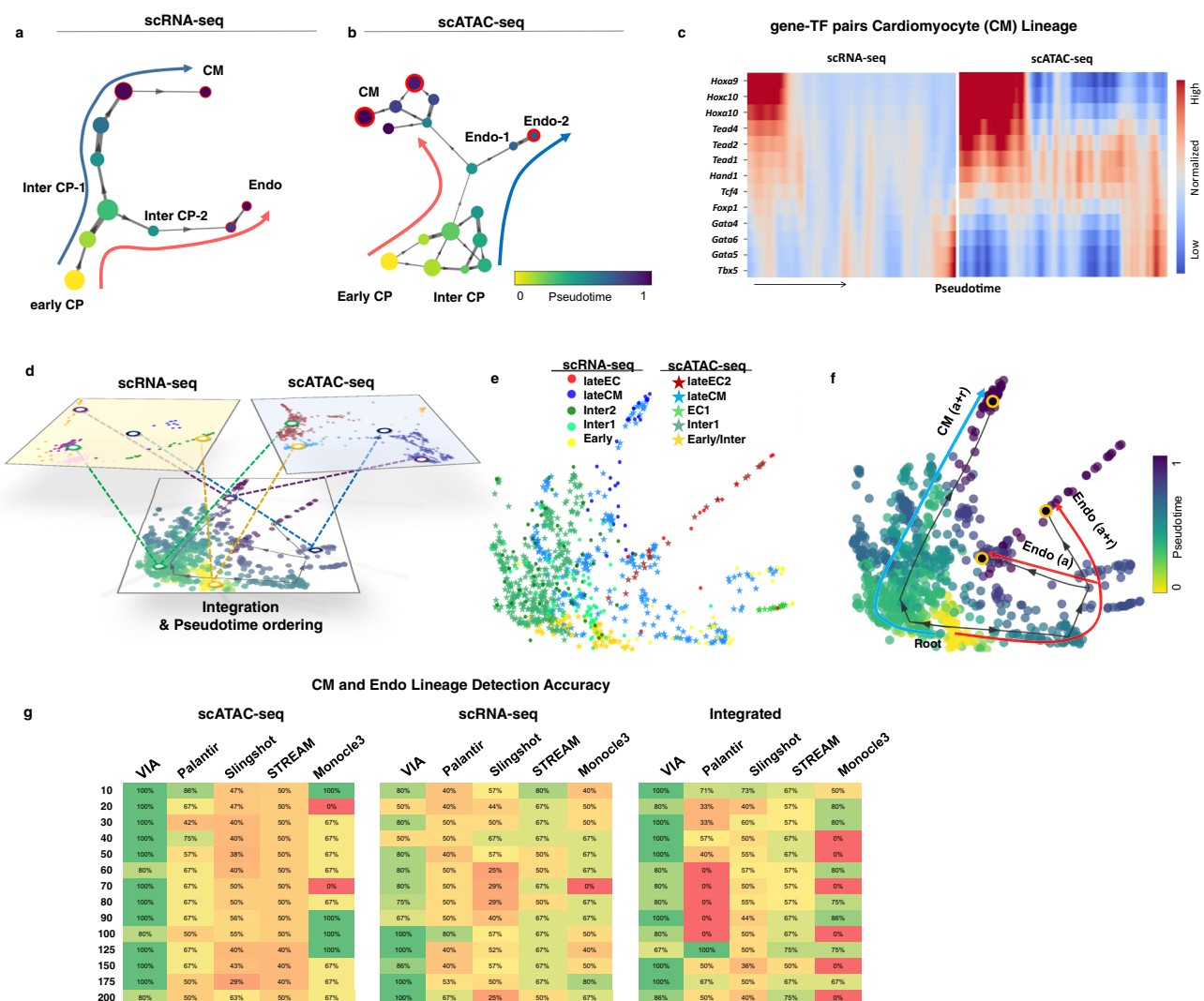

**Fig. 5 Multi-omic integrated analysis of scRNA-seq and scATAC-seq cardiac progenitors. a** VIA graph for scRNA-seq data only and (**b**) scATAC-seq data only. **c** Gene-TF pair expression along VIA inferred pseudotime for each CM lineage (see Supplementary Fig. 19 for Top five most differentially expressed genes for each VIA node along each lineage as well as node-level TF motif accessibility) (**d**) schematic of data integration of the individual sc-modalities (**e**) scRNA-seq and scATAC-seq data of Isl1+ Cardiac Progenitors (CPs) integrated using Seurat3 before PHATE. Colored by annotated cell-type, and experimental modality (**f**) Colored by VIA pseudotime with VIA-inferred trajectory towards Endothelial and Myocyte lineages projected on top (**g**) Accuracy of detecting the CM and Endo lineages in the individual and integrated data. This is challenging for other methods which either detect several early/intermediate stages or merge cell fates (see outputs for these methods in Supplementary Figs. 20, 21).

More importantly, VIA automatically generates a pseudotem-poral ordering of relevant cells (without requiring manual selection of relevant cells as done in Jia et al.[26]) along each lineage and their marker-TF pairs (Fig. 5c and Supplementary Fig. 19f for differential gene expression analysis). Hence, VIA can be used to faithfully interpret relationships between transcription factor dynamics and gene expression in an unsupervised manner. The highlighted gene and TF pairs in the cardiac lineage show a strong correlation between expression and accessibility of *Gata* and Homeobox *Hox* genes which are known to be related to the regulation of cardiomyocyte proliferation[29–31]. VIA's reliable performance against user-reconfiguration (number of PCs, individual or integrated omic data) suggests its utility in transferable interpretation between scRNA-seq and scATAC-seq data.

**VIA preserves global connectivity when scaling to millions of cells.** VIA is designed to be highly scalable and offers automated lineage prediction without extensive dimension reduction or subsampling even at large cell counts. To showcase this, we use VIA to explore the 1.3-million scRNA-seq mouse organogenesis cell atlas (MOCA)[7]. While this dataset is inaccessible to most TI methods from a runtime and memory perspective, VIA can efficiently resolve the underlying developmental heterogeneity, including nine major trajectories (Fig. 6a, b) with a runtime of ~40 min, compared to the next fastest method PAGA which has a runtime of 3 h, Palantir and STREAM which takes over 4 and 6.5 h respectively. Other methods like Slingshot and CellRank were deemed infeasible due to extremely long runtimes on much smaller datasets. (Supplementary Table 3 for a summary of runtimes). Going beyond the computational efficiency, VIA also preserves wider neighborhood information and reveals a globally connected topology of MOCA which is otherwise lost in the Monocle3 analysis which first reduces the input data dimensionality using UMAP.

The overall cluster graph of VIA consists of three main branches that concur with the known developmental process at early organogenesis[32] (Fig. 6a). It starts from the root stem which has

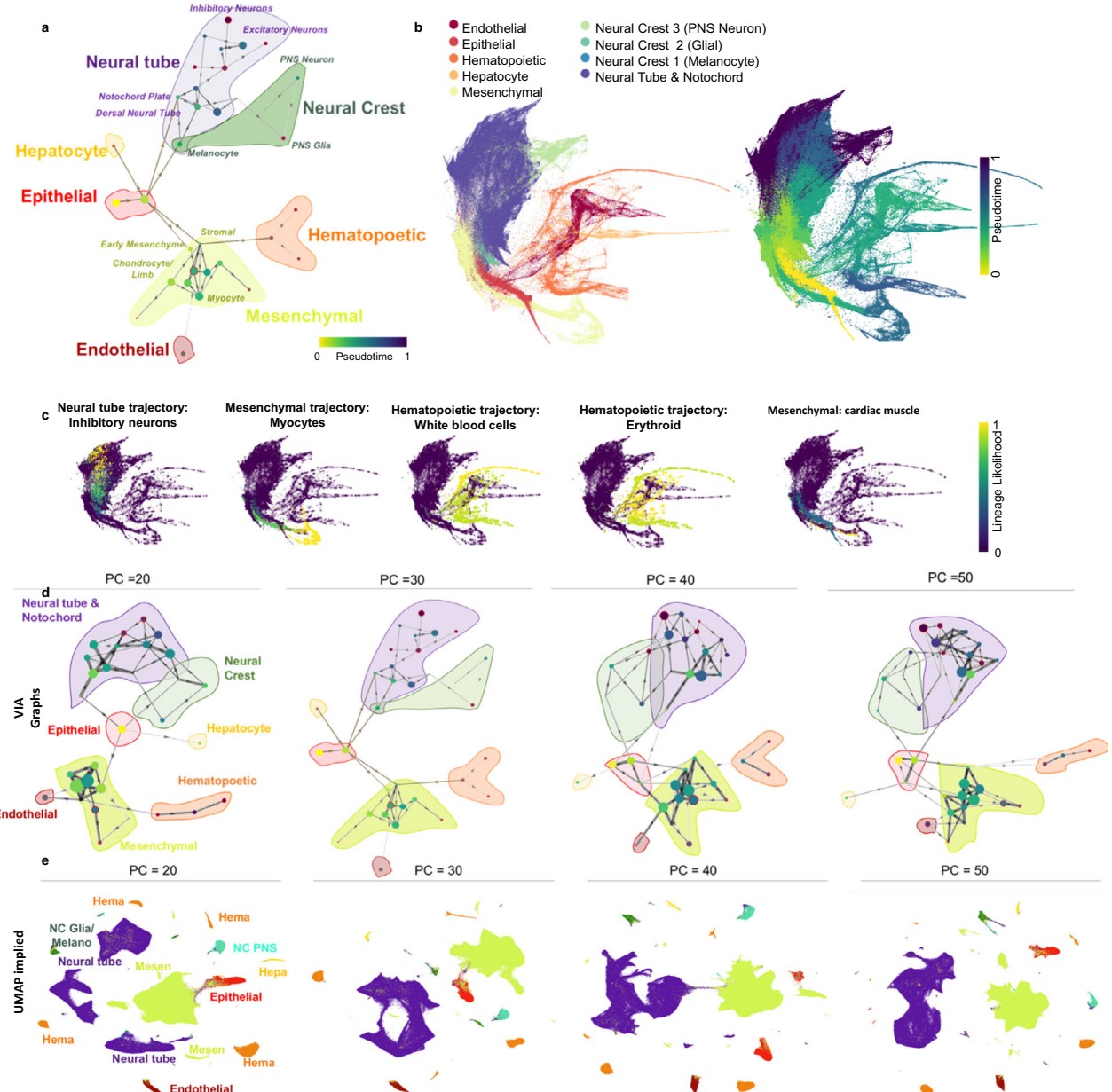

**Fig. 6 VIA accurately infers global connectivity and sub-trajectories in the 1.3-million scRNA-seq mouse organogenesis cell atlas. a** MOCA graph trajectory (nodes colored by pseudotime) and shaded-colored regions corresponding to major cell groups. Stem branch consists of epithelial cells derived from ectoderm and endoderm, leading to two main branches: (1) the mesenchymal and (2) the neural tube and neural crest. Other major groups are placed in the biologically relevant neighborhoods, such as the adjacencies between hepatocyte and epithelial trajectories; the neural crest and the neural tube; as well as the links between early mesenchyme with both the hematopoietic cells and the endothelial cells (see Supplementary Note 7). **b** Colored by VIA pseudotime. **c** Lineage pathways and probabilities of neuronal, myocyte and WBC lineages. **d** VIA graph preserves key relationships across choice of number of PCs, whereas (**e**) UMAP embedding is first step in Monocle3 and highly susceptible to choice of number of PCs (or K in KNN see Fig. 22).

a high concentration of E9.5 early epithelial cells made of multiple sub-trajectories (e.g., epidermis, and foregut/hindgut epithelial cells derived from the ectoderm and endoderm). The stem is connected to two distinct lineages: (1) mesenchymal cells originated from the mesoderm which arises from interactions between the ectoderm and endoderm[27,28,32,33–35] and (2) neural tube/crest cells derived from neurulation when the ectoderm folds inwards[34].

The sparsity of early cells (only ~8% are E9.5) and the absence of earlier ancestral cells make it particularly challenging to capture the simultaneous development of trajectories. However, VIA is able to capture the overall pseudotime structure depicting early organogenesis (Fig. 6b). For instance, at the junction of the epithelial-to-mesenchymal branch, we find early mesenchymal cells from

E9.5–E10.5. Cells from later mesenchymal developmental stages (e.g., myocytes from E12.5–E13.5) reside at the leaves of the branch. Similarly, at the junction of epithelial-to-neural tube, we find dorsal tube neural cells and notochord plate cells which are predominantly from E9.5–E10.5 and more developed neural cells at branch tips (e.g., excitatory and inhibitory neurons appearing at E12.5–E13.5). In contrast, the pseudotime gradient of PAGA's nodes offer little salient information at this scale, with 90% of cells predicted to be in the first 10% of the pseudotime color scale (see Supplementary Fig. 22c, d).

VIA also consistently places the other smaller dispersed groups of trajectories (e.g., endothelial, hematopoietic) in biologically relevant neighborhoods (see Supplementary Note 7 for a detailed explanation of VIA's structural connections supported by known

transitions in organogenesis literature). While VIA's connected topology offers a coarse-grained holistic view, it does not compromise the ability to delineate individual lineage pathways, such as the erythroid and white blood cell lineages within the hematopoietic super group (consistent with annotations made by Cao et al.[7]) as shown in Fig. 6c.

As such, TI using VIA uniquely preserves both the global and local structures of the data. Whilst manifold-learning methods are often used to extensively reduce dimensionality to mitigate the computational burden of large single-cell datasets, they tend to incur loss of global information and be sensitive to input parameters. VIA is sufficiently scalable to bypass such a step, and therefore retains a higher degree of neighborhood information when mapping large datasets. This is in contrast to Monocle3's[7] UMAP-reduced inputs that reveal different disconnected super-groups and fluctuating connectivity depending on input parameters. As shown in Fig. 6e (and Supplementary Fig. 22 for varying KNN), methods such as Monocle3 which require a very low dimensional representation (e.g., first 2-3 components of UMAP) for TI are susceptible to unpredictable changes in the composition of super cell groups, their relative positions and inter-connectivity. For instance, in UMAP, the neural tube group is sometimes shown as a single super group, and other times fragmented across the embedding without context of neighboring groups. Similarly the hematopoietic supergroup is shown as a single, two or even three separate groups dispersed across the embedding landscape (Fig. 6e). In contrast, VIA uncovers biologically consistent structures across the same range of parameters. In VIA, the cells belonging to these fine-grained supergroups remain connected and neighborhood relationships are preserved, for instance the neural crest cells (containing Peripheral Nervous System neurons and glial cells) remain adjacent to the neural tube (Figs. 6d, Supplementary Fig. 22a).

**VIA's lazy-teleporting MCMCs delineate mesoderm differentiation in mass cytometry data.** Broad applicability of TI beyond transcriptomic analysis is increasingly critical, but existing methods have limitations contending with the disparity in the data structure (e.g., sparsity and dimensionality) across a variety of single-cell data types. While we have shown that VIA can be used to successfully interrogate scATACseq, scRNAseq, and their integrated data, we further investigate whether VIA can cope with the significant drop in data dimensionality (10–100), as often presented in flow/mass cytometry data, and still delineate continuous biological processes.

We applied VIA on a time-series mass cytometry data (28 antibodies, 90K cells) capturing murine ESCs differentiation toward mesoderm cells[36]. The mESCs are captured at 12 intervals within the first 11 days and hence provide sufficiently granular temporal annotation to allow a correlation assessment of the inferred pseudotimes. We quantified that the pseudotimes computed by VIA shows a Pearson correlation of ~88% with the actual annotated days. We further verified that VIA's performance is critically improved by the lazy-teleporting MCMCs (Fig. 7d), without which the correlation drops closer to PAGA's. Palantir and Monocle3 suffer from low connectivity of cells between the Day 0–1 and the subsequent early stages (finding disconnected trajectories even when increasing K in KNN), and thus result in loss of pseudotime gradient and low correlation to the true annotations.

More importantly, unlike previous analysis[36] of the same data which required chronological labels to visualize the chronological developmental hierarchy, we ran VIA without such supervised adjustments and accurately captured the sequential development. Not only can it achieve faster runtime (running in 2 min on the full antibody-feature set versus Slingshot which required 6 h even on a subset of first 5 PCs see Supplementary Table 3 for more runtime comparisons), VIA detects three terminal states corresponding to cells in the final developmental stages of Day 10–11 which are indicated by upregulation of *Pdgfra*, *Cd44* and *Gata4* mesodermal markers (Fig. 7e). In contrast, other methods struggle to identify the correct terminal states (e.g., Palantir, STREAM and Slingshot Fig. 7f) and do not depict salient structures (e.g., STREAM where the Day 10–11 branch is placed in between Day 0 and Day 5 branches).

**VIA captures morphological trends of live cells in cell cycle progression.** Apart from the omics technologies, optical microscopy is a powerful parallel advance in single-cell analysis for generating the "fingerprint" profiles of cell morphology. Such spatial information is typically obscured in sequencing data, but can effectively underpin cell states and functions without costly and time-consuming sequencing protocols. However, trajectory predictions based on morphological profiles of single cells have only been scarcely studied until recently, but advancements in high-throughput imaging cytometry are now making large-scale image data generation and related studies feasible. We thus sought to test if VIA can predict biologically relevant progress based on single-cell morphological snapshots captured by our recently developed high-throughput imaging flow cytometer, called FACED[13]—a technology that is at least 100 times faster than state-of-the-art imaging flow cytometry (IFC) (Fig. 8a).

Our FACED imaging platform captured multiple image contrasts of single cells, including FL, and quantitative phase images (QPI), which measure high-resolution biophysical properties of cells, which are otherwise inaccessible in other methods[37]. Using the QPIs captured by FACED, we first generated spatially-resolved single-cell biophysical profiles of two live breast cancer cell types (MDA-MB231 and MCF7) undergoing cell cycle progressions (38 features including cell shape, size, dry mass density, optical density and their subcellular textures (see Supplementary Tables 6, 7 for definitions of features)). The QPI together with the FL images of individual cells were also used to train a convolutional neural network-based regression model for predicting the DNA content. We first validated that there is a high correlation (Pearson's correlation coefficient $r = 0.72$) between the actual DNA content determined by the FL images and DNA content predicted by the QPI (Supplementary Fig. 24a). In addition, the predicted percentages of cells in each cell cycle phases (i.e., G1, S and G2/M) by the biophysical profile are highly consistent with the ground truth defined by the DNA dye (Supplementary Fig. 24b). Based on the biophysical profiles as validated by the above tests, VIA reliably reconstructed the continuous cell-cycle progressions from G1-S-G2/M phase of both types of live breast cancer cells (Methods) (Fig. 8b–g).

Intriguingly, according to the pseudotime ordered by VIA, not only does it reveal the known cell growth in size and mass[38], and general conservation of cell mass density[39] (as derived from the FACED images (Methods)) throughout the G1/S/G2 phases, but also a slow-down trend during the G1/S transition in both cell types, consistent with the lower protein-accumulation rate during S phase[40] (Fig. 8f, g). The variation in biophysical textures (e.g., peak phase, and phase fiber radial distribution) along the VIA pseudotime likely relates to known architectural changes of chromosomes and cytoskeletons during the cell cycles (Fig. 8f, g). We find other methods on this dataset to be sensitive to the choice of early cells and detecting intermediate cells as terminal cell fates (e.g., Palantir, Slingshot), and often adding additional edges or branches (e.g., STREAM, PAGA), see Supplementary Fig. 23 for Palantir, Slingshot, Monocle3, STREAM and PAGA

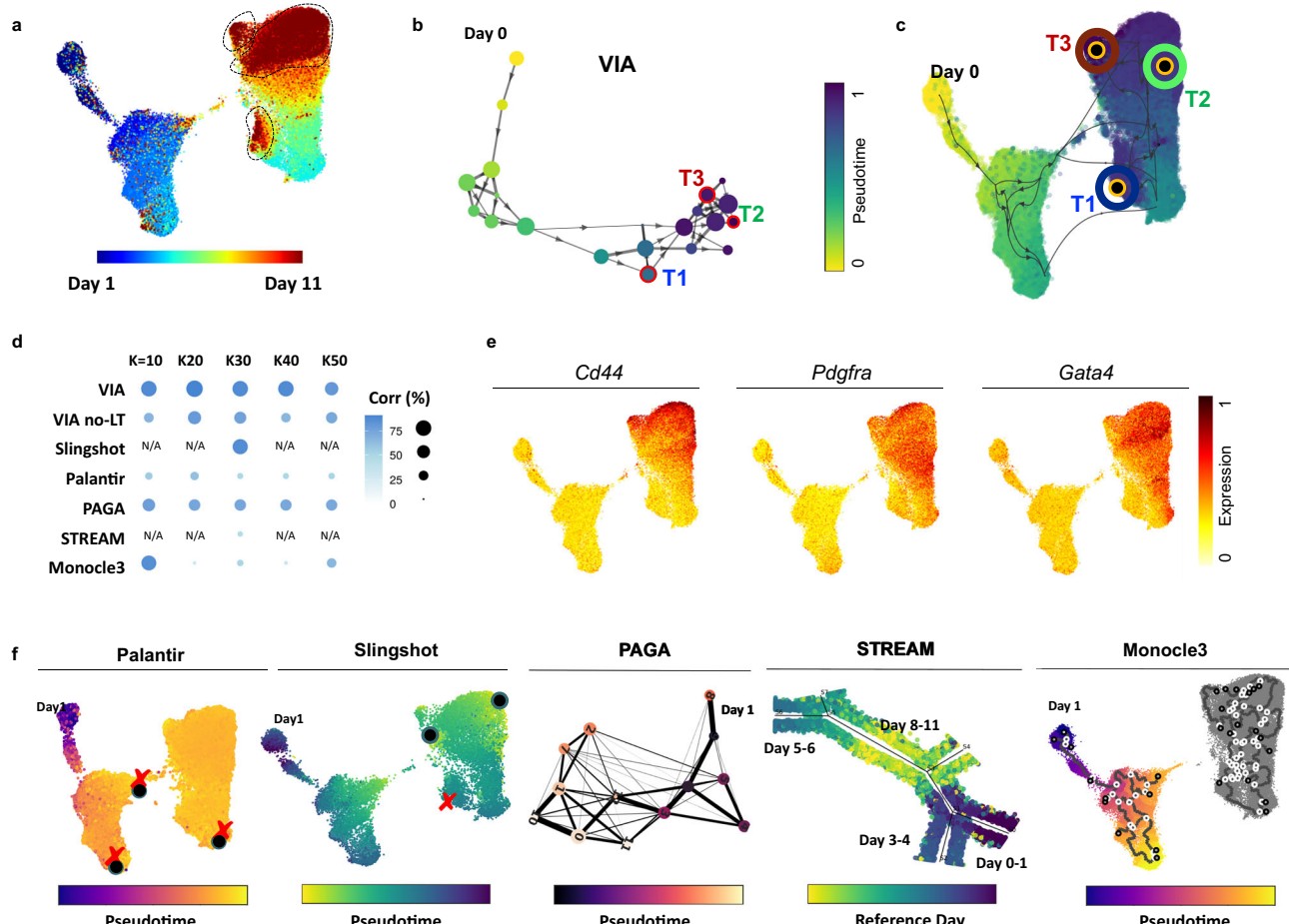

**Fig. 7 VIA analysis of mESC differentiation toward mesoderm cells from mass cytometry. a** UMAP plot colored by annotated days 0–11. Three regions of Day 10–11 marked in dotted black lines. **b** VIA cluster-graph colored by pseudotime. **c** Terminal states and VIA output projected onto UMAP. Terminal states are located in the areas containing Day 10–11 cells. **d** Comparison of Pearson correlation of pseudotime and annotated Days across TI methods for varying number of K number of nearest neighbors. PAGA and Palantir's pseudotime computation is misguided by the weak link connecting Day 0 cells to other early cells. The effect is that Day 0 cells appear exaggeratedly far, while the remaining early and late cells are temporally squeezed. VIA's 2-step pseudotime computation produces a pseudotime scale closer to the annotated dates. "VIA no-LT" denotes VIA without the lazy-teleporting MCMC stage of the pseudotime calculation. For Slingshot and STREAM there is no K (NN) setting thus only a single correlation value is presented. STREAM's pseudotime is distorted by the insertion of Day 8–11 cells in between Day 0 and Day 5. **e** Gene expression of key mesodermal markers. **f** Example outputs of Palantir, PAGA and Slingshot with the terminal states (black circles) predicted by Slingshot and Palantir. Red "X" denotes incorrect (false positive) or missing (false negative) terminal state. STREAM places Day 10–11 cells in between Day 0 and Day 5–6 cells.

outputs. The slowdown during the S-phase is missed by the gene trend prediction available in other methods. To probe subsets of the morphological features, we remove volume and volume related features (e.g., Dry Mass, Area) and test whether this can still be used to infer the topology and cell ordering that reveals the slow-down observed in the S-phase. We found that VIA is consistently able to reveal these trends in both cell lines, whereas other methods struggle to maintain the linear progression expected along the cell-cycle with spurious linkages emerging (see Supplementary Figs. 25, 26) and intermediate states being selected as final G2 stages. These results further substantiate the growing body of work[41–44] on imaging biophysical cytometry for gaining a mechanistic understanding of biological systems, especially when combined with omics analysis[45].

## Discussion

With the growing scale and complexity of single-cell datasets, there is an unmet need for accurate cell fate prediction and lineage detection in complex topologies manifested in biology (not limited to trees). This challenge, broadly faced by the current

TI methods, is compounded by susceptibility to algorithmic parameter changes, limited scalability to large data size; and insufficient generalizability to multi-omic data beyond transcriptomic data. We introduced VIA, which alleviates these challenges by fast and scalable construction of cluster-graph of cells, followed by pseudotime, and reconstructing cell lineages based on lazy-teleporting random walks and MCMC simulations. This strategy critically relaxes common constraints on graph traversal and causality that impede accurate prediction of elusive lineages and less populous cell fates. We validated the efficacy of these measures in terms of detecting various challenging topologies on simulated data, as well as robust prediction of cell fates and temporally changing feature trends on a variety biological processes (spanning epigenomic, transcriptomic, integrated omic, as well as imaging and mass cytometric data) to show that VIA detects pertinent biological lineages and their pathways that remain undetected by other methods.

Notably, VIA distinguished between dendritic subtypes in an scRNA-seq hematopoiesis dataset; identified the rare delta cell islet in pancreatic development, a population requiring manual

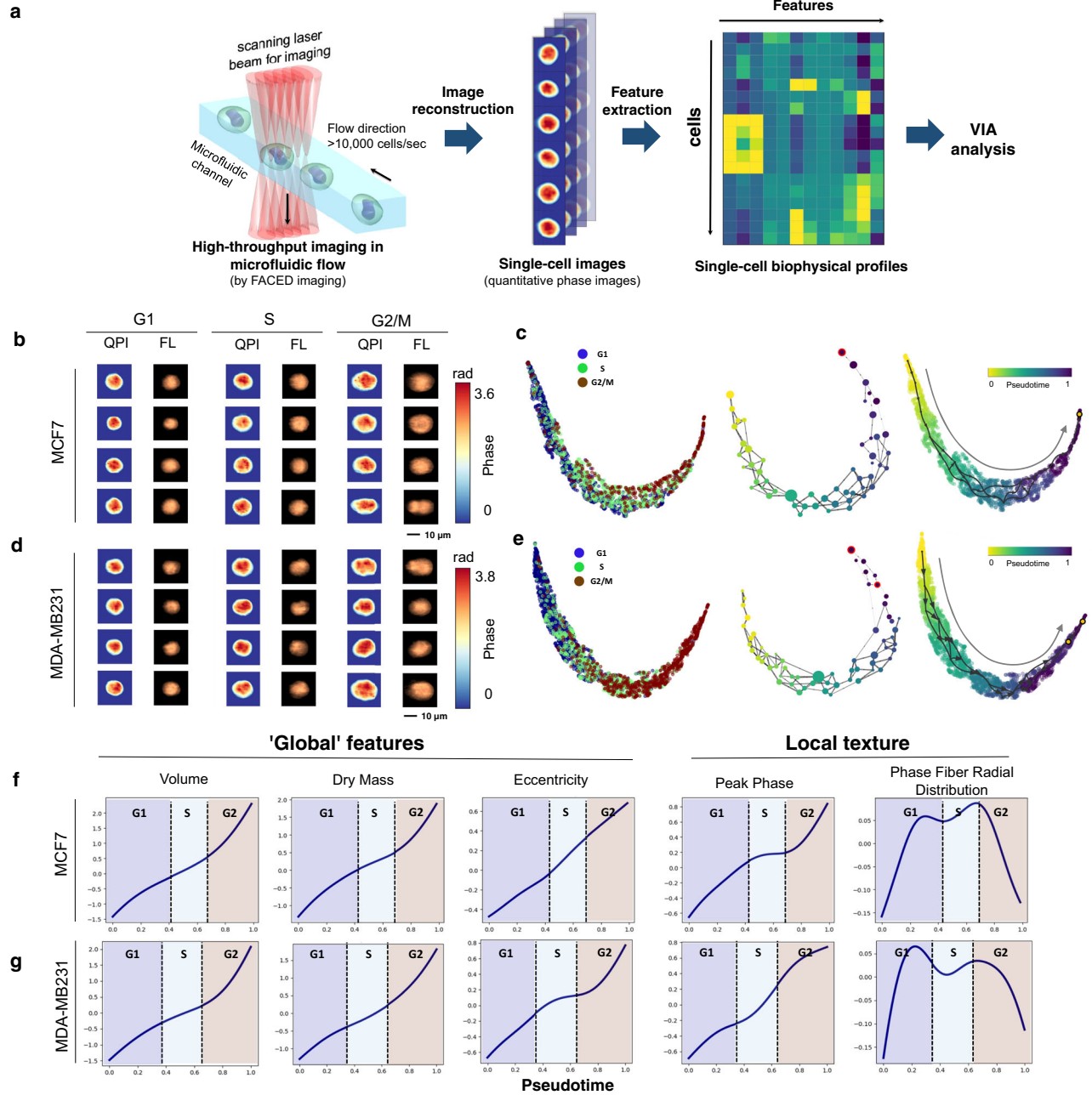

**Fig. 8 VIA predicts cell cycle progression based on single-cell biophysical morphology. a** FACED high-throughput imaging flow cytometry of MDA-MB231 and MCF7 cells, followed by image reconstruction and biophysical feature extraction. See "Methods" detailed experimental workflow. **b** Randomly sampled quantitative phase images (QPI) and fluorescence images (FL) of MCF7 cells and (**d**) MDA-MB231 cells. **c** Single-cell UMAP embedding colored by the known cell-cycle phase (left), given by DNA-labeled fluorescence images. VIA inferred cluster-graph topology, nodes colored by pseudotime (mid) and UMAP colored by VIA pseudotime for MCF7. **d-e** VIA analysis repeated for MDA-MB231 cells. **f** Unsupervised image-feature-trends of global and local biophysical textures against VIA pseudotime for MCF7 and (**g**) MDA-MB231 cells (see Supplementary Table 6 for feature definitions). Cell cycle pseudotime boundaries are defined here as the intersection of the pseudotime probability density functions of each cell cycle stage (annotated based on fluorescence intensity).

assignment in other TI methods; and revealed the bifurcation towards cardiomyocyte and endothelial lineage commitment in a multi-omic scATAC-seq and scRNA-seq dataset which proved challenging for other methods. In order to demonstrate that these biological findings are robust to user parameter tuning, we conducted a series of 'stress tests' of the inferred topology and cell fates on both simulated and biological data, which show that VIA behaves more predictably (allowing controllable degrees of analytical granularity) and accurately than other methods with

regards to topology and lineage prediction. In other methods, user parameter choice can incur fragmentation or spurious linkages in the modeled topology, and consequently only yield biologically sensible lineages for a narrow sweet spot of parameters (see the summary in Supplementary Fig. 1 and sample outputs by other methods in Supplementary Figs. 6, 9, 11–13, 15–18, 20–21, 22, and 23).

We also demonstrated on the 1.3 million MOCA dataset that VIA is highly scalable with a runtime of ~40 min (compared to

3–4 h on the next fastest method). Importantly, VIA not only recovers the fine-grained sub-trajectories, but also maintains global connectivity between related cell types and thus captures key relationships among lineages in early embryogenesis. It also computes a more salient pseudotime measure supported by lazy-teleporting MCMCs, compared to other methods whose pseudotime scale was distorted at such high cell counts (Supplementary Fig. 22c, d). We showed that methods which require UMAP (or t-SNE) before parsing MOCA are highly susceptible to user defined input parameters that can significantly and unpredictably fragment the global topology.

We also assessed whether VIA can be generalized to non-transcriptomic single-cell datasets, especially those with significant dimensionality disparity compared to sequencing data. We first applied VIA to the mESC CyTOF dataset and showed that the lazy-teleporting MCMCs strategy in VIA enables it to outperform other methods in correctly correlating the pseudotime of the mesoderm development to the annotated dates. We finally explored the utility of VIA in analyzing emerging image-based single-cell biophysical profile data. We showed that VIA not only successfully identified the progression of G1/S/G2 stages, but also revealed the subtle changes in biophysical-related cellular properties, which are otherwise obscured in other methods. VIA could thus motivate new strategies in single-cell analysis that link cellular biophysical phenotypes and biochemical/biomolecular information, to discover how molecular signatures translate into the emergent cellular biophysical properties, which has already shown effective in studies of cancer, ageing, and drug responses. Overall, VIA offers an advancement to TI methods to robustly study a diverse range of single-cell data. Together with its scalable computation and efficient runtime, VIA could be useful for multifaceted exploratory analysis to uncover biological processes, potentially those deviated from the healthy trajectories

## Methods

**VIA algorithm**. VIA applies a scalable probabilistic method to infer cell state dynamics and differentiation hierarchies by organizing cells into trajectories along a pseudotime axis in a nearest-neighbor graph which is the basis for subsequent random walks. Single cells are represented by graph nodes that are connected based on their feature similarity, e.g., gene expression, transcription factor accessibility motif, protein expression, or morphological features of cell images. A typical routine in VIA mainly consists of four steps:

*Accelerated and scalable cluster-graph construction.* VIA first represents the single-cell data in a k-nearest-neighbor (KNN) graph where each node is a cluster of single cells. The clusters are computed by our recently developed clustering algorithm, PARC[14]. In brief, PARC is built on hierarchical navigable small world[46] accelerated KNN graph construction and a fast community-detection algorithm (Leiden method[47]), which is further refined by data-driven pruning. The combination of these steps enables PARC to outperform other clustering algorithms in computational run-time, scalability in data size and dimension (without relying on subsampling of large-scale, high-dimensional single-cell data (>1 million cells)), and sensitivity of rare-cell detection. We employ the cluster-level topology, instead of a single-cell-level graph, for TI as it provides a coarser but clearer view of the key linkages and pathways of the underlying cell dynamics without imposing constraints on the graph edges. Together with the strength of PARC in clustering scalability and sensitivity, this step critically allows VIA to faithfully reveal complex topologies namely cyclic, disconnected and multifurcating trajectories (Fig. 2). If the user prefers to use another clustering method or group-labels of cell types according to apriori information, VIA can easily accommodate such a substitution and the robustness of the lazy-teleporting random walks to different clustering approaches is shown in Supplementary Note 6 and Figs. S30–32 for real and synthetic data. The root cell is initialized by the user in one of two ways: If for instance there are some cell type/group/cluster level labels available in advance, the desired starting group can be indicated to VIA, which will then automatically select a cluster in its cluster-graph that contains a majority of this particular cell type/group classification. In the case of many clusters satisfying this criteria, it subsequently proceeds to select the cluster in the VIA graph that has connectivity metrics indicative of a root (leaf) node (such as high out degree, low betweenness and low centrality). The user can also choose to provide a specific single cell as the root node. In the case that the user wishes to select the root based on the VIA

graph, one would save the VIA-cluster-graph labels and use them to guide selection of the root node as described in the first approach.

*Probabilistic pseudotime computation.* The trajectories are then modeled in VIA as: (i) lazy-teleporting random walk paths along which the pseudotime is computed and further refined by (ii) MCMC simulations. The root is a single cell chosen by the user. These two sub-steps are detailed as follows:

## Lazy-teleporting random walk

We first compute the pseudotime as the expected hitting time of a *lazy-teleporting* random walk on an undirected cluster-graph generated in Step 1. The lazy-teleporting nature of this random walk ensures that as the sample size grows, the expected hitting time of each node does not converge to the stationary probability given by local node properties, but instead continues to incorporate the wider global neighborhood information[15]. Here we highlight the derivation of the closed form expression of the hitting time of this modified random walk with a detailed derivation in Supplementary Note 2.

The cluster graph constructed in VIA is defined as a weighted connected graph $\mathbf{G}$ ($V$, $E$, $W$) with a vertex set $V$ of $n$ vertices (or nodes), i.e., $V = \{v_1, \cdots, v_n\}$ and an edge set $E$, i.e., a set of ordered pairs of distinct nodes. $W$ is an $n \times n$ weight matrix that describes a set of edge weights between node $i$ and $j$, $w_{ij} \geq 0$ are assigned to the edges $(v_i, v_j)$. For an undirected graph, $w_{ij} = w_{ji}$, the $n \times n$ probability transition matrix, $P$, of a standard random walk on G is given by

$$\mathbf{P} = \mathbf{D}^{-1}\mathbf{W} \tag{1}$$

where $D$ is the $n \times n$ degree matrix, which is a diagonal matrix of the weighted sum of the degree of each node, i.e., the matrix elements are expressed as

$$d_{ij} \begin{cases} \sum_k w_{ik} & , i = j \\ 0 & , i \neq j \end{cases} \tag{2}$$

where $k$ are the neighboring nodes connected to node $i$. Hence, $d_{ii}$ (which can be reduced as $d_i$) is the degree of node $i$. We next consider a *lazy* random walk, defined as $Z$, with probability $(1 - x)$ of being lazy (where $0 < x < 1$), i.e., staying at the same node, then

$$\mathbf{Z} = x\mathbf{P} + (1 - x)\mathbf{I} \tag{3}$$

where $I$ is the identity matrix. When teleportation occurs with a probability $(1 - \alpha)$, the modified lazy-teleporting random walk $Z'$ can be written as follows, where $J$ is an $n \times n$ matrix of ones.

$$\mathbf{Z}' = \alpha\mathbf{Z} + (1 - \alpha)\frac{1}{n}\mathbf{J} \tag{4}$$

Here we adapt the concept of personalized PageRank vector, originally used for recording (or *ranking*) personal preferences of a web-surfer toward particular website pages[48], to *rank* the importance of other nodes (clusters of cells) to a given node, depending on the similarities among nodes (related to $P$ in the graph), and the lazy-teleporting random walk characteristics in the graph (set by probabilities of teleporting and being lazy). Based on this concept, one could model the likelihood to transit from one node (cluster of cells) to another, and thus construct the pseudotime based on the hitting time, which is a parameter describing the expected number of steps it takes for a random walk that starts at node $i$ and visit node $j$ for the first time. Consider the teleporting probability of $(1 - \alpha)$ and a seed vector $s$ specifying the initial probability distribution across the $n$ nodes (such that $\sum_m s_m = 1$, where $s_m$ is the probability of starting at node $m$) the personalized PageRank vector $pr_\alpha(s)$ (which is defined as a column vector) is the unique solution to[49]

$$\mathbf{pr}_\alpha(\mathbf{s})^{\mathbf{T}} = \alpha\mathbf{pr}_\alpha(\mathbf{s})^{\mathbf{T}}\mathbf{Z} + (1 - \alpha)\mathbf{s}^{\mathbf{T}} \tag{5}$$

Substituting $Z$ (Eq. (3)) into Eq. (5), we can express the personalized PageRank vector $pr_\alpha(s)$ in terms of the inverse of the $\beta$-normalized Laplacian, $R_{\beta,NL}$ of the modified random walk (Supplementary Note 2), i.e.,

$$\mathbf{pr}_\alpha(\mathbf{s})^{\mathbf{T}} = \beta\mathbf{s}^{\mathbf{T}}\mathbf{D}^{-0.5}\mathbf{R}_{\beta,\mathbf{NL}}\mathbf{D}^{0.5} \tag{6}$$

where $\beta = \frac{2(1-\alpha)}{(2-\alpha)}$, and $\mathbf{R}_{\beta,\mathbf{NL}} = \sum_{m=1} \frac{\Phi_m \Phi_m^{\mathbf{T}}}{[\beta + 2x(1-\beta)\eta_m]}$, $\Phi_m$ and $\eta_m$ are the m$^{th}$ eigenvector and eigenvalue of the normalized Laplacian. In the expression of $\mathbf{R}_{\beta,\mathbf{NL}}$, the $\beta$ and $x$ regulate the weight of contribution in each eigenvalue-eigenvector pair of the summation such that the first eigenvalue-eigenvector pair (corresponding to the stationary distribution and given by the local-node degree-properties) remains included in the overall expression, but does not overwhelm the global information provided by subsequent "eigen-pairs". Moreover, computation of $\mathbf{R}_{\beta,\mathbf{NL}}$ is not limited to a subset of the first $k$ eigenvectors (bypassing the need for the user to select a suitable threshold or subset of eigenvectors) since the dimensionality is not on the order of number of cells, but equal to the number of clusters and hence all eigenvalue–eigenvector pairs can be incorporated without causing a bottleneck in runtime.

The expected hitting time from node $q$ to node $r$ is given by[50],

$$h_\alpha(q, r) = \frac{[\mathbf{pr}_\alpha(\mathbf{e_r})^{\mathbf{T}}](\mathbf{r})}{d_r} - \frac{[\mathbf{pr}_\alpha(\mathbf{e_r})^{\mathbf{T}}](\mathbf{q})}{d_q} \tag{7}$$

where $e_i$ is an indicator vector with 1 in the $i^{th}$ entry and 0 elsewhere (i.e., $s_m = 1$ if $m = i$ and $s_m = 0$ if $m \neq i$). We can substitute Eq. (6) into Eq. (7), making use of the

fact that $\frac{1}{d_r} = [\mathbf{D}^{-1}\mathbf{e_r}](r)$, and $\mathbf{D}^{-0.5}\mathbf{R}_{\beta,\mathbf{NL}}\mathbf{D}^{-0.5}$ is symmetric, to obtain a closed form expression of the hitting time in terms of $\mathbf{R}_{\beta,\mathbf{NL}}$

$$h_\alpha(q,r) = \beta(\mathbf{e_r} - \mathbf{e_q})^T\mathbf{D}^{-0.5}\mathbf{R}_{\beta,\mathbf{NL}}\mathbf{D}^{-0.5}\mathbf{e_r} \qquad (8)$$

## MCMC simulation

The hitting time metric computed in Step-1 is used to infer graph-directionality. Instead of pruning edges in the "reverse" direction, edge-weights are biased based on the time difference between nodes using the logistic function with growth factor b = 1.

$$f(t) = \frac{1}{1 + e^{-b(t_0 - t_1)}} \qquad (9)$$

We then recompute the pseudotimes on the forward biased graph: Since there is no closed form solution of hitting times on a *directed* graph, we perform MCMC simulations (parallely processed to enable fast simulations of 1000 s of teleporting, lazy random walks starting at the root node of the cluster graph) and use the first quartile of the simulated pseudotime values for a respective node as the refined pseudotime for that node relative to the root. This refinement step ensures that the pseudotime is robust to the spurious links (or conversely, links that are too weakly weighted) that can distort calculations based purely on the closed form solution of hitting times (Supplementary Fig. 7d). By using this 2-step pseudotime computation, VIA mitigates the issues of convergence issues and spurious edge-weights, both of which are common in random-walk pseudotime computation on large and complex datasets[15].

*Automated terminal-state detection.* The algorithm uses the refined directed and weighted graph (edges are re-weighted using the refined pseudotimes) to predict which nodes represent the terminal states based on a consensus vote of pseudotime and multiple vertex connectivity properties, including out-degree (i.e., the number of edges directed out of a node), closeness $C(q)$, and betweenness $B(q)$.

$$C(q) = \frac{1}{\sum_{q \neq r} l(q,r)} \qquad (10)$$

$$B(q) = \sum_{r \neq q \neq t} \frac{\sigma_{rt}(q)}{\sigma_{rt}} \qquad (11)$$

$l(q,r)$ is the distance between node $q$ and node $r$ (i.e., the sum of edges in a shortest path connecting them). $\sigma_{rt}$ is the total number of shortest paths from node $r$ to node $t$. $\sigma_{rt}(q)$ is the number of these paths passing through node $q$. The consensus vote is performed on nodes that score above (or below for out-degree) the median in terms of connectivity properties. We show on multiple simulated and real biological datasets that VIA more accurately predicts the terminal states, across a range of input data dimensions and key algorithm parameters, than other methods attempting the same (Supplementary Fig. 1).

*Automated trajectory reconstruction.* VIA then identifies the most likely path of each lineage by computing the likelihood of a node traversing towards a particular terminal state (e.g., differentiation). These lineage likelihoods are computed as the visitation frequency under lazy-teleporting MCMC simulations from the root to a particular terminal state, i.e., the probability of node $i$ reaching terminal-state $j$ as the number of times cell $i$ is visited along a successful path (i.e., terminal-state $j$ is reached) divided by the number of times cell $i$ is visited along all of the simulations. In contrast to other trajectory reconstruction methods which compute the shortest paths between root and terminal node[1,2], the lazy-teleporting MCMC simulations in VIA offer a probabilistic view of pathways under relaxed conditions that are not only restricted to the random-walk along a tree-like graph, but can also be generalizable to other types of topologies, such as cyclic or connected/disconnected paths. In the same vein, we avoid confining the graph to an absorbing Markov chain[18,51] (AMC) as this places prematurely strict/potentially inaccurate constraints on node-to-node mobility and can impede sensitivity to cell fates (as demonstrated by VIA's superior cell fate detection across numerous datasets (Supplementary Fig. 1).

**Downstream visualization and analysis.** VIA generates a visualization that combines the network topology and single-cell level pseudotime/lineage probability properties onto an embedding based on UMAP or PHATE. Generalized additive models (GAMs) are used to draw edges found in the high-dimensional graph onto the lower dimensional visualization (Fig. 1). An unsupervised downstream analysis of cell features (e.g., marker gene expression, protein expression or image phenotype) along pseudotime for each lineage is performed (Fig. 1). Specifically, VIA plots the expression of features across pseudotime for each lineage by using the lineage likelihood properties to weight the GAMs. A cluster-level lineage pathway is automatically produced by VIA to visualize heat maps at the cluster-level along a lineage-path to see the regulation of genes. VIA provides the option of gene imputation before plotting the lineage specific gene trends. The imputation is fast as it relies on the single-cell KNN (scKNN) graph computed in Step 1. Using an affinity-based imputation method[52], this step computes a "diffused" transition matrix on the scKNN graph used to impute and denoise the original gene expressions.

**Simulated data.** We employed the DynToy[5] (https://github.com/dynverse/dyntoy) package, which generates synthetic single-cell gene expression data (~1000 cells × 1000 "genes"), to simulate different complex trajectory models. Using these datasets, we tested that VIA consistently and more accurately captures both tree and non-tree like structures compared to other methods (Fig. 2). The types of topologies span multifurcating, cyclic, connected (hybrid of cyclic and multifurcating) and disconnected (hybrid of the first three). All methods are subject to the same data preprocessing steps, PCA dimension reduction, and root-cell to initialize the path.

The composite accuracy metric assesses multiple layers of the inferred trajectory, taking into account the topological similarity between the reference model and the inferred topology, the correlation between the real and "pseudo" times, and the prediction accuracy of the terminal cell fates (lineages). Absolute measurements of similarities are converted into a percentage scale before taking the arithmetic mean (of the 5 metrics, see below) which gives the composite accuracy. Since PAGA does not predict lineages, the composite score is simply the average of the first 4 metrics for PAGA. A detailed explanation of the 5 metrics can be referred to Supplementary Note 3. The 5 metrics are:

*Ipsen–Mikhailov (IM).* is used to measure the similarity of global graph topology. The IM ranges from 0 to 1 and equals the difference in spectral densities of two graphs.

*Graph edit distance (GED).* is the cost of converting $G_{TI}$ to $G_{REF}$ with the least possible number of operations. Each operation has a cost of one and includes insertion/deletion of edges *and* nodes.

*F1-branch score.* We compute the harmonic mean of recall and precision for the local branch accuracy relative to the reference model. A False Negative edge in the inferred model is when there is an edge in the reference model between cell types that is absent in the inferred trajectory. A False Positive edge in the inferred model is an edge that is not actually present in the reference model.

*Temporal correlation.* Pearson correlation coefficient is used as a measure of how closely the inferred pseudotime follows the true sampling times.

*F1-cell fate score.* Similar to the F1-branch score, we use the harmonic mean of recall and precision to quantify the prediction accuracy of terminal states.

**Benchmarked methods.** The methods were mainly chosen based on their superior performance in a recent large-scale benchmarking study[5], including a select few recent methods claiming to supersede those in the study. Specifically, recent and popular methods exhibiting reasonable scalability, and automated cell fate prediction in multi-lineage trajectories, not limited to tree-topologies, were favored as candidates for benchmarking (see Supplementary Table 1 for the key characteristics of methods). Performance stress-tests in terms of lineage detection of each biological dataset, automated gene trend prediction along lineages, and pseudotime correlation were conducted over a range of key input parameters (e.g., numbers of k-nearest neighbors, highly variable genes (HVGs), PCs) and preprocessing protocols (see Supplementary Fig. 1). Methods that focus exclusively on a single data modality or on topology without predicting cell fates and their lineage pathways (e.g., TinGa[53], Tempora[54]) were generally not included in the benchmarking as they would require manual selection of cell fates and differentiation pathways. All comparisons were run on a computer with an Intel(R) Xeon (R) W-2123 central processing unit (3.60 GHz, 8 cores) and 126 GB RAM.

Details of parameter settings for each of the benchmarked methods can be found in Supplementary Tables 4, 5, with an emphasis on the rationale for changes deviating from default parameters.

Quantifying terminal state prediction accuracy for parameter tests was done using the F1-score, defined as the harmonic mean of recall and precision and calculated as:

$$F_1 = \frac{tp}{tp + 0.5(fp + fn)} \qquad (12)$$

Where $tp$ is a true-positive: the identification of a terminal cluster that is in fact a final differentiated cell fate; $fp$ is a false positive identification of a cluster as terminal when in fact it represents an intermediate state; and $fn$ is a false negative where a known cell fate fails to be identified.

Downstream analysis enabled by the automated lineage prediction capabilities of each method is key to facilitating the exploration of biological data. The unsupervised gene-trend analysis inferred by VIA is compared to the lineage gene-trends predicted by other methods both quantitatively and qualitatively. We follow an approach used by Chen et al.[3] where pseudotime is correlated against expression of a marker gene known to monotonically increase along the lineage. The gene-expression of such markers can be considered a surrogate for the correct sampling time and thus the resulting correlation is an indication of the accuracy of cell ordering by pseudotime. We also provide a side-by-side comparison of the predicted topology and gene-trends generated by each method to visually assess how well separated the predicted lineages are (e.g., if multiple lineages that represent distinct

cell fates exhibit significant cross-talk in the plotted trends or uniquely express the genes most relevant to their lineages). The Pearson correlation coefficient is given by $\rho_{x,y}$, where $\sigma_X$ is the standard deviation and $\mu_X$ is the mean of X

$$\rho_{x,y} = \frac{E[(X - \mu_X) - (Y - \mu_Y)]}{\sigma_X \sigma_Y} \tag{13}$$

Built-in functions for gene-trend plotting (wherever available), and in other cases manually selection of branches/clusters or extension of a method by adding GAMs to general gene-trend curves was required to facilitate comparison (e.g., PAGA and STREAM). Additionally, when methods cannot automatically detect all the relevant lineages, we either chose the most relevant lineage (e.g., for the megakaryocyte lineage, we plotted its CD41 marker gene along the detected erythroid lineage which often absorbed the smaller megakaryocytic cell line), or we noted that the lineage was missed, (e.g., in the small delta cell population in the endocrine dataset) when the lost lineage was not an obvious part of another lineage. Given that these nuances are not necessarily captured by the correlation coefficient, the outputs of the gene-trend plots inferred by each method are shown for three datasets which have multiple lineages of different abundances, and well known lineage markers (scRNA-seq and scATAC-seq hematopoiesis, and endocrine genesis in Supplementary Figs. 9, 11, 15).

*PAGA[55].* It uses a cluster-graph representation to capture the underlying topology. PAGA computes a unified pseudotime by averaging the single-cell level diffusion pseudotime computed by DPT, but requires manual specification of terminal cell fates and clusters that contribute to lineages of interest in order to compare gene expression trends across lineages.

*Palantir[2].* It uses diffusion-map[56]. components to represent the underlying trajectory. Pseudotimes are computed as the shortest path along a KNN-graph constructed in a low-dimensional diffusion component space, with edges weighted such that the distance between nodes corresponds to the diffusion pseudotime[57] (DPT). Terminal states are identified as extrema of the diffusion maps that are also outliers of the stationary distribution. The lineage-likelihood probabilities are computed using Absorbing Markov Chains (constructed by removing outgoing edges of terminal states, and thresholding reverse edges).

*Slingshot[1].* It is designed to process low-dimensional embeddings of the single-cell data. By default Slingshot runs clustering based on Gaussian mixture modeling and recommends using the first few PCs as input. Slingshot connects the clusters using a minimum spanning tree and then fits principle curves for each detected branch. It uses the orthogonal projection against each principal curve to fit a separate pseudotime for each lineage, and hence the gene expressions cannot be compared across lineages. Also, the runtimes are prohibitively long for large datasets or high input dimensions.

*CellRank[13].* This method combines the information of RNA velocity (computed using scVelo[58]) and gene-expression to infer trajectories. Given it is mainly suited for the scRNA-seq data, with the RNA-velocity computation limiting the overall runtime for larger dataset, we limit our comparison to the pancreatic dataset which the authors of CellRank used to highlight its performance.

*Monocle3[4].* The workflow consists of three steps: the first is to project the data to two or three dimensions using UMAP (this is a strict requirement), followed by Louvain clustering on a K-Nearest Neighbor graph constructed in the low-dimensional UMAP space. A cluster-graph is then created and partitioned to deduce disconnected trajectories. Subsequently, it learns a principal graph in the low-dimensional space along which it calculates pseudotimes as the geodesic distance from root to cell.

*STREAM[3].* After selecting the desired number of PCs, STREAM projects the cells to a lower dimensional PCA space using a non-linear dimensionality reduction method (such as Modified Locally Linear Embedding, Spectral Embedding or UMAP). In the embedded space, STREAM constructs a tree-model trajectory using an Elastic Principal Graph implementation called ElPiGraph. The results are visualized as a branching structure or re-organized as a subway plot relative to a user-designated starting branch.

**Biological data**. The preprocessing steps described below for each dataset are not included in the reported runtimes as these steps are typically very fast, (typically <1–10% of the total runtime depending on the method. E.g., only a few minutes for pre-processing 100,000 s of cells) and only need to be performed once as they remain the same for all subsequent analyses. It should also be noted that visualization (e.g., UMAP, t-SNE) are not included in the runtimes. VIA provides a subsampling option at the visualization stage to accelerate this process for large datasets without impacting the previous computational steps. However, to ensure fair comparisons between TI methods (e.g., other methods do not have an option to compute the embedding on a subsampled input and transfer the results between the full trajectory and the sampled visualization, or rely on a slow version of tSNE),

we simply provide each TI method with a precomputed visualization embedding on which the computed results are projected.

*ScRNA-seq of mouse pre-B cells.* This dataset[21] models the pre-BI cell (Hardy fraction C′) process during which cells progress to the pre-BII stage and B cell progenitors undergo growth arrest and differentiation. Measurements were obtained at 0, 2, 6, 12, 18, and 24 h (h) for a total of 313 cells × 9075 genes. We follow a standard Scanpy preprocessing recipe[59] that filters cells with low counts, and genes that occur in <3 cells. The filtered cells are normalized by library size and log transformed. The top 5000 HVG are retained. Cells are renormalized by library count and scaled to unit variance and zero mean. VIA identifies the terminal state at 18–24 h and accurately recapitulates the gene expression trends[21] along inferred pseudotime of *IgII1, Slc7a5, Fox01, Myc, Ldha,* and *Lig4.* (Supplementary Fig. 6a). We show the results generalize across a range of PCs for two values of K of the graph with higher accuracy in locating the later cell fates than Slingshot and Palantir (Supplementary Fig. 6b).

*ScRNA-seq of human CD34+ bone marrow cells.* This is a scRNA-seq dataset of 5800 cells representing human hematopoiesis[2]. We used the filtered, normalized and log-transformed count matrix provided by Setty et al.[2] with PCA performed on all the remaining (~14,000) genes. The cells were annotated using SingleR[60]. which automatically labeled cells based on the hematopoietic reference dataset Novershtern Hematopoietic Cell Data—GSE24759[61]. The annotations are in agreement with the labels inferred by Setty et al. for the seven clusters, including the root HSCs cluster that differentiates into six different lineages: monocytes, erythrocytes, and B cells, as well as the less populous megakaryocytes, cDCs, and pDCs. VIA consistently identifies these lineages across a wider range of input parameters and data dimensions (e.g., the number of K and PCs provided as input to the algorithms see Fig. 2p, and Supplementary Figs. 7–9). Notably, the upregulated gene expression trends of the small populations can be recovered in VIA, i.e., pDC and cDC show elevated CD123 and CSF1R levels relative to other lineages, and the upregulated CD41 expression in megakaryocytes (Supplementary Figs. 7–9).

*ScRNA-seq of human embryoid body.* This is a midsized scRNA-seq dataset of 16,825 human cells in embryoid bodies (EBs)[17]. We followed the same preprocessing steps as Moon et al. to filter out dead cells and those with too high or low library count. Cells are normalized by library count followed by square root transform. Finally the transformed counts are scaled to unit variance and zero mean. The filtered data contained 16,825 cells × 17,580 genes. PCA is performed on the processed data before running each TI method. VIA identifies six cell fates, which, based on the upregulation of marker genes as cells proceed towards respective lineages, are in accord with the annotations given by Moon et al. (see the gene heatmap and changes in gene expression along respective lineage trajectories in Supplementary Fig. 13). Note that Palantir and Slingshot do not capture the cardiac cell fate, and Slingshot also misses the neural crest (see the F1-scores summary for terminal state detection Supplementary Fig. 13).

*ScRNA-seq of mouse organogenesis cell atlas.* This is a large and complex scRNA-seq dataset of mouse organogenesis cell atlas (MOCA) consisting of 1.3 million cells[4]. The dataset contains cells from 61 embryos spanning 5 developmental stages from early organogenesis (E9.5–E10.5) to organogenesis (E13.5). Of the 2 million cells profiled, 1.3 million are "high-quality" cells that are analysed by VIA. The runtime is ~40 min which is in stark contrast to the next fastest tool Palantir which takes 4 h (excluding visualization). The authors of MOCA manually annotated 38 cell-types based on the differentially expressed genes of the clusters. In general, each cell type exclusively falls under one of 10 major and disjoint trajectories inferred by applying Monocle3 to the UMAP of MOCA. The authors attributed the disconnected nature of the ten trajectories to the paucity of earlier stage common predecessor cells. We followed the same steps as Cao et al.[4] to retain high-quality cells (i.e., remove cells with less than 400 mRNA, and remove doublet cells and cells from doubled derived sub-clusters). PCA was applied to the top 2000 HVGs with the top 30 PCs selected for analysis. VIA analyzed the data in the high-dimension PC space. We bypass the step in Monocle3 which applies UMAP on the PCs prior to TI as this incurs an additional bias from choice of manifold-learning parameters and a further loss in neighborhood information. As a result, VIA produces a more connected structure with linkages between some of the major cell types that become segregated in UMAP (and hence Monocle3), and favors a biologically relevant interpretation (Fig. 2, Supplementary Fig. 11). A detailed explanation of these connections (graph-edges) extending between certain major groups using references to literature on organogenesis is presented in Supplementary Note 3.

*ScRNA-seq of murine endocrine development[23].* This is an scRNA-seq dataset of E15.5 murine pancreatic cells spanning all developmental stages from an initial endocrine progenitor-precursor (EP) state (low level of *Ngn3,* or *Ngn3[low]*), to the intermediate EP (high level of *Ngn3,* or *Ngn3[high]*) and Fev+ states, to the terminal states of hormone-producing alpha, beta, epsilon, and delta cells[23]. Following steps by Lange et al.[18] we preprocessed the data using scVelo to filter genes, normalize each cell by total counts over all genes, keep the top most variable genes, and take the log-transform. PCA was applied to the processed gene matrix. We assessed the performance of VIA and other TI methods (CellRank, Palantir, Slingshot) across a range of number of retained HVGs and input PCs (Fig. 2m, Supplementary Figs. 16, 16, 18, 29).

*ScATAC-seq of human bone marrow cells*. This scATAC-seq data profiles 3072 cells isolated from human bone marrow using FL activated cell sorting (FACS), yielding 9 populations[20]: HSC, MPP, CMP, CLP, LMPP, GMP, MEP, mono, and plasmacytoid DCs (Fig. 3a and Supplementary Figs. 10, 11). We examined TI results for two different preprocessing pipelines to gauge how robust VIA is on the scATAC-seq analysis which is known to be challenging due to its extreme intrinsic sparsity. We used the pre-processed data consisting of PCA applied to the z-scores of the transcription factor (TF) motifs used by Buenrostro et a[20]. Their approach corrects for batch effects in select populations and weighting of PCs based on reference populations and hence involves manual curation. We also employed a more general approach used by Chen et al.[22] which employs ChromVAR to compute k-mer accessibility z-scores across cells. VIA infers the correct trajectories and the terminal cell fates for both of these inputs, again across a wide range of input parameters (Fig. 3d and Supplementary Figs. 11–13).

*ScRNA-seq and scATAC-seq of* Isl1+ *cardiac progenitor cells*. This time-series dataset captures murine *Isl1*+ cardiac progenitor cells (CPCs) from E7.5 to E9.5 characterized by scRNA-seq (197 cells) and scATAC-seq (695 cells)[26]. The *Isl1*+ CPCs are known to undergo multipotent differentiation to cardiomyocytes or endothelial cells. For the scRNA-seq data, the quality filtered genes and the size-factor normalized expression values are provided by Jia et al.[26] as a "Single Cell Expression Set" object in R. Similarly, the cells in the scATAC-seq experiment were provided in a "SingleCellExperiment" object with low quality cells excluded from further analysis. The accessibility of peaks was transformed to a binary representation as input for TF-IDF (term frequency-inverse document frequency) weighting prior to singular value decomposition (SVD). The highlighted TF motifs in the heatmap (Fig. 2j) correspond to those highlighted by Jia et al. We tested the performance when varying the number of SVDs used. We also considered the outcome when merging the scATAC-seq and scRNA-seq data using Seurat3[62]. Despite the relatively low cell count of both datasets, and the relatively underrepresented scRNA-seq cell count, the two datasets overlapped reasonably well and allowed us to infer the expected lineages in an unsupervised manner (Fig. 2d and Supplementary Fig. 8. In contrast, Jia et al. performed a supervised TI by manually selecting cells relevant to the different lineages (for the scATAC-seq cells) and choosing the two diffusion components that best characterize the developmental trajectories in low dimension[26].

*Mass cytometry data of mouse embryonic stem cells (mESC)*. This is a mass cytometry (or CyTOF) dataset, consisting of 90,000 cells and 28 antibodies (corresponding to ~7000 cells each from Day 0–11 measurements), that represents differentiation of mESC to mesoderm cells[36]. An arcsinh transform with a scaling factor of five was applied on all features—a standard procedure for CyTOF datasets, followed by normalization to unit variance and zero mean. All 28 antibodies are used by the TI methods (with the exception of Slingshot which requires PCA followed by subsetting of the first 5 PCs in order to computationally handle the high cell count) (Supplementary Fig. 9). To improve Palantir performance we used 5000 waypoints (instead of default 1200) but this takes almost 20 min to complete (excluding time taken for embedding the visualization). VIA runs in ~3 min and produces results consistent with the known ordering and identifies regions of Day 10–11 cells.

*Single-cell biophysical phenotypes derived from imaging flow cytometry*. This is the in-house dataset of single-cell biophysical phenotypes of two different human breast cancer types (MDA-MB231 and MCF7). Following our recent image-based biophysical phenotyping strategy[63,64] we defined the spatially-resolved biophysical features of a cell in a hierarchical manner based on both bright-field and QPI captured by the FACED imaging flow cytometer (i.e., from the bulk features to the subcellular textures)[65]. At the bulk level, we extracted the cell size, dry mass density, and cell shape. At the subcellular texture level, we parameterized the global and local textural characteristics of optical density and mass density at both the coarse and fine scales (e.g., local variation of mass density, its higher-order statistics, phase entropy radial distribution etc.). This hierarchical phenotyping approach[63,64] allowed us to establish a single-cell biophysical profile of 38 features, which were normalized based on the z-score (see Supplementary Tables 4, 5). All these features, without any PCA, are used as input to VIA. In order to weigh the features, we use a mutual information classifier to rank the features, based on the integrated FL intensity of the FL FACED images of the cells (which serve as the ground truth of the cell-cycle stages). Following normalization, the top three features (which relate to cell size) are weighted (using a factor between 3 and 10).

### Imaging flow cytometry experiment
*FACED imaging flow cytometer setup*. A multimodal FACED IFC platform was used to obtain the quantitative phase and FL images of single cells in microfluidic flow at an imaging throughput of ~70,000 cells/s. The light source consisted of an Nd:YVO picosecond laser (center wavelength = 1064 nm, Time-Bandwidth) and a periodically-poled lithium niobate (PPLN) crystal (Covesion) for second harmonic generation of a green pulsed beam (center wavelength = 532 nm) with a repetition rate of 20 MHz. The beam was then directed to the FACED module, which mainly consists of a pair of almost-parallel plane mirrors. This module generated a linear array of 50 beamlets (foci) which were projected by an objective lens (40X, 0.6NA, MRH08430, Nikon) on the flowing cells in the microfluidic channel for imaging. Each beamlet was designed to have a time delay of 1 ns with the neighboring beamlet

in order to minimize the FL crosstalk due to the FL decay. Detailed configuration of the FACED module can be referred to Wu et al.[13]. The epi-fluorescence image signal was collected by the same objective lens and directed through a band-pass dichroic beamsplitter (center: 575 nm, bandwidth: 15 nm). The filtered orange FL signal was collected by the photomultiplier tube (PMT) (rise time: 0.57 ns, Hamamatsu). On the other hand, the transmitted light through the cell was collected by another objective lens (40X, 0.8NA, MRD07420, Nikon). The light was then split equally by the 50:50 beamsplitter into two paths, each of which encodes different phase-gradient image contrasts of the same cell (a concept similar to Scherlien photography[66]). The two beams are combined, time-interleaved, and directed to the photodetector (PD) (bandwidth: >10 GHz, Alphalas) for detection. The signals obtained from both PMT and PD were then passed to a real-time high-bandwidth digitizer (20 GHz, 80 GS/s, Lecroy) for data recording.

*Cell culture and preparation*. MDA-MB231 (ATCC) and MCF7 (ATCC), which are two different breast cancer cell lines, were used for the cell cycle study. The culture medium for MDA-MB231was ATCC modified RPMI 1640 (Gibco) supplemented with 10% fetal bovine serum (FBS) (Gibco) and 1% antibiotic-antimycotic (Anti-Anti) (Gibco), while that for MCF7 was DMEM supplemented with 10% FBS (Gibco) and 1% Anti-Anti (Gibco). The cells were cultured inside an incubator under 5% $CO_2$ and 37 °C, and subcultured twice a week. 1e6 cells were pipetted out from each cell line and stained with Vybrant DyeCycle orange stain (Invitrogen).

**Reporting summary**. Further information on research design is available in the Nature Research Reporting Summary linked to this article.

### Data availability
The Pancreatic data used in this study are available in the Gene Expression Omnibus (GEO) database under accession code GSE132188. The Cardiac progenitor data used in this study are available in the ENA repository under the accession code PRJEB23303 or from [https://github.com/loosolab/cardiac-progenitors]. The B-cell data used in this study are available in the STATegraData GitHub repository. [https://github.com/STATegraData/STATegraData] and under the GEO database under accession code GSE75417. The Mass cytometry mesoderm data used in this study are available in the Cytobank database [https://community.cytobank.org/cytobank/experiments/71953]. The scRNA-seq Human Hematopoiesis data used in this study are available in the Human Cell Atlas data portal database [https://data.humancellatlas.org/explore/projects/091cf39b-01bc-42e5-9437-f419a66c8a45]. The Embryoid Body data used in this study are available in the Mendeley Data database at [https://doi.org/10.17632/v6n743h5ng.1]. The Mouse organogenesis data used in this study are available in the NCBI Gene Expression Omnibus database under accession code GSE119945. The FACED cell cycle data used in this study are available at https://github.com/ShobiStassen/VIA and on FigShare database [https://doi.org/10.6084/m9.figshare.13601405.v1]. The scATAC-seq Hematopoiesis data used in the GEO database under accession code GSE96772. Processed scATAC-seq data, which include PC values and TF scores per cell can be found in Data S1. [https://doi.org/10.1016/j.cell.2018.03.074]. The Toy Data used in the study is available in [https://zenodo.org/record/5205377] https://doi.org/10.5281/zenodo.5205377 [https://github.com/ShobiStassen/VIA].

### Code availability
VIA is available as a pip installable python library "pyVIA" with tutorials and sample data available on https://github.com/ShobiStassen/VIA, https://pypi.org/project/pyVIA/ and https://zenodo.org/record/5205377[67].

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

## Acknowledgements

This work was funded by the Research Grants Council of the Hong Kong Special Administrative Region of China (grant nos. 17208918, 17209017, 17259316, RFS2021-7S06, and C7047-16G).

## Author contributions

K.K.T., J.W.K.H. and S.V.S. conceived the project. S.V.S developed the algorithm and software to analyze the data. S.V.S. G.G.K.Y. and K.K.Y.W. designed and performed the FACED experiments. K.K.T. and S.V.S. wrote the paper. All authors commented on and edited the text.

## Competing interests

The authors declare no competing interests.

**Additional information**

