## [Peer Review File · Nature Communications]

Generalized and scalable trajectory inference in single-cell omics data with VIAReviewers' Comments:

Reviewer #1:

Remarks to the Author:

This manuscript proposed a trajectory inference method VIA, which is a graph-based approach utilizing lazy-teleporting random walks with MCMC refinement. In contrast to the single-cell graph based random walk methods, like DPT, the random walks of VIA are based on the cluster-graph so that it can reduce the time for pseudo-time inference and improve its robustness to different parameter settings. VIA does not make any assumptions on the trajectory structures so that it is flexible to characterize variety of topologies. Besides, the case studies of VIA on several real single-cell omics datasets along with validation of in-situ fluorescence image capture also valid the analysis. Overall, the paper is well written and provides sufficient supplementary details. I have the following questions and suggestions about the method and its applications to improve the manuscript.

VIA has been shown to be effective for rare branch detection, which implies, on the other side, it may also be sensitive to the data noises. How does its performance change with the increasing noise level of the data?

For graph edge accuracy, which is computed based on an F1-score. I think a false negative is: there exists an edge to connect to cell types that are not connected in the reference. Besides, since you compared VIA with Monocle3, PAGA, Slingshot..., what are the parameter settings of these algorithms? As the comparisons with other algorithms, you are supposed to search for many other parameters which may affect the performance apart from number of pcs and KNN. In the other experiments, this suggestion also holds true.

In Fig2.b, when you varied K in (KNN), the Graph edge accuracy of mnoncle3 is not changed. I wonder whether it is a true or not? Can you explain this?

In Fig3, if you would like to compare VIA with other algorithms under different number of pcs and KNN, a grid search of all the integers between 20-200 and 10-70 is more reasonable. Also, since the cluster are detected by PARC, I am interested in that whether the robustness to the number of pcs is mainly because of PARC, can you further confirm this?

In the experiment of E15.5 murine pancreatic cells, you said that 'VIA detects small endocrine Delta lineages and Beta subtypes'. Which part of the algorithm lead to this? PARC, or the lazy-teleporting MCMC?

For Fig7, since Slingshot is an algorithm that orders each single cell on the inferred cell lineage, while VIA just infers the trajectory of cell clusters. It seems unfair to compare these two algorithms considering time consuming.

VIA first represents the single-cell data in a k-nearest-neighbor (KNN) graph where each node is a cluster of single cells. However, VIA cannot infer the trajectories of single cells. Since the cell fate decision and changes of gene expression in single cells (especially the cells around bifurcation) are more important. Can the lazy-teleporting MCMC be applied to single cells in the PARC clusters to derive single cell trajectory?

Minor comments:

The 2D embedding plots of VIA and benchmarking methods in Figure 2 seem to be colored by different ways and the color legends are not displayed, which may be confusing.

The sizes in the figure texts are not unified and some of them are too small to read.

Fig1, caption, line84, 'can remain (orange arrows)' -> 'can remain Lazy (orange arrows)'

Fig2.a, M6 is missing on the first graph.

Reviewer #2:

Remarks to the Author:

The manuscript describes the development of a trajectory inference of large single cell datasets generated for example by scRNA-seq, imaging mass cytometry or imaging flow cytometry. There are a

plethora of algorithms that been designed to do exactly this task and the manuscript demonstrates advantages over a selection of these algorithms using a range of toy and real world data.

However this is a fast moving field and new algorithms appear almost monthly which appear to be capable of solving the same problems this approach is designed to be optimised for. For example how does this algorithm compare with

STREAM (Chen et.al. Nat. Comms, 2019) - which also compares performance with the algorithms benchmarked here and others such as scTDA, Wishbone, TSCAN, SLICER, DPT, GPFates, Mpath, SCUBA which are not benchmarked in this work.

TinGa (Todorov et. al., BioInofrmatics 2020) - which appears to work well for the cycle and more complex geometries.

Tempora (Tran et.al. PLOS COMPUTATIONAL BIOLOGY 2020)

and these are to name just a few.

Also an example is given of the analysis of the cell cycle progress of cells using Imaging Flow Cytometry. However this type of analysis has been carried out previously (Blasi et.al. 2016, Eulenburg 2017 - Nat Comms) and the trajectory is not very complex so a simple diffusion map approach can work here.

The authors also use an ergodic approach (Kafri et.al.2013 Nature) to infer details about certain cell characteristics, however as Kafri discussed, care must be take to not use any predictor measure corrected to the response variable i.e. if you are inferring dry mass, volume then no correlated variables (e.g. cell radius, area etc) should be used in the pseudo time alignment.

I believe the manuscript is probably better suited to a method type journal as it appear to be an addition to an already large set algorithms to do this job and also the details of some of the examples need to be further elucidated as discussed above.

Reviewer #3:

Remarks to the Author:

1) This paper presents the VIA method for inferring trajectories in single-cell data, regardless of the - omics type. It relies on one metric, the F1-score (that assesses how accurately cell fates were identified by each method), to assess the accuracy of the VIA method and compare it to 5 state-of-the-art methods. This F1-score seems too weak of a metric to me, as it doesn't capture the structure of the returned trajectory. I can think of different trajectories (successively bifurcating, trifurcating, or even disconnected), that could have exactly the same cell fates, and thus return the same F1-score. I would thus recommend adding a second metric, next to the F1-score, that would capture the topology of the different trajectories. Moreover, the F1-score metric that was chosen by the authors doesn't allow them to compare all the tools that they included in the study (as PAGA doesn't automatically detect terminal states). If the paper chose to use only one metric to compare all methods, this metric should at least allow to compare all methods. In my opinion, the paper would benefit from adding at least one more metric to compare the methods on.

2) Small comment related to the previous one: in figure 2a), the authors state that the F1-score can't be computed for PAGA as this method doesn't automatically return cell fates. However, an F1-score was then computed for PAGA for the cyclic dataset (presented figure 2b), and disconnected dataset (presented in figure 2c). Please clarify in the text, in the methods, or in the description of figure 2 how and why the F1-score was computed for some datasets but not for others. The fact that the F1-scores

presented on the right of figure 2 correspond to a different set of methods for each dataset is confusing.

3) It is unclear to me why, in each figure where VIA is compared to other methods, it is not being compared to all the other methods. For example, why did the authors choose not to include the results of PAGA in figure 4? Or Monocle3 in figure 7?

4) The paragraph starting at line 378 should, in my opinion, be re-phrased. Historically, the first trajectory inference (TI) tools were designed to be applied on cytometry data (Wanderlust, Monocle). This type of data contains a much better ratio of cells over features than scRNA-seq data, and suffers less from sparsity, which typically makes it easier to find trajectories in this type of data. However, the way the paragraph is written now, the authors seem to suggest that most TI methods can only handle transcriptomic data, and that handling cytometry data would be a new advantage of VIA. I would thus advise to rephrase this paragraph to avoid misleading the reader.

5) The authors emphasize the fact that VIA doesn't require any user input parameters, compared to other methods. Only when reading the methods section do we discover that VIA needs the user to define a starting point to run. This should be stated earlier in the paper in my opinion (around line 25 for instance), as this is an information that any potential user of the method will be interested in. Side comment: as PARC allows a fast visualisation of the data, I'm wondering if allowing the user to choose a starting point in this representation of the data would be a nice way to simplify the choice of a starting point for the user.

6) The authors highlight the fact that the VIA method's computational advantage allows it to process larger amounts of dimensions, avoiding information loss due to dimensionality reduction. However, they test VIA on all original genes only in one dataset (presented in figure 4), and apply VIA on PCs for the other datasets. It would be interesting to see how the method performs on the original dimensions and how it compares to other methods on the synthetic datasets, and in figure 3.

7) Comment related to the previous one: it is surprising to see that VIA performs optimally when applied on > 6000 HVGs in figure 4, while Slingshot performs best on less than 2000 HVGs (which is more expected). This might be due to the fact that the F1-score captures only the accuracy of the end states recovered by the different methods, and doesn't represent differences in topology. It would be interesting to see whether the same increase in accuracy is observed when using more HVGs in the synthetic data and in figure 3.

REVIEWER #1 (Expertise: Trajectory inference, single cell data):

*This manuscript proposed a trajectory inference method VIA, which is a graph-based approach utilizing*
*lazy-teleporting random walks with MCMC refinement. In contrast to the single-cell graph based random*
*walk methods, like DPT, the random walks of VIA are based on the cluster-graph so that it can reduce the*
*time for pseudo-time inference and improve its robustness to different parameter settings. VIA does not*
*make any assumptions on the trajectory structures so that it is flexible to characterize variety of*
*topologies. Besides, the case studies of VIA on several real single-cell omics datasets along with*
*validation of in-situ fluorescence image capture also valid the analysis. Overall, the paper is well written*
*and provides sufficient supplementary details. I have the following questions and suggestions about the*
*method and its applications to improve the manuscript.*

*R1.Q1 VIA has been shown to be effective for rare branch detection, which implies, on the other side, it*
*may also be sensitive to the data noises. How does its performance change with the increasing noise level*
*of the data?*

**R1.A1** We thank the reviewer for this question and have investigated the impact of noise on VIA's
performance. We first wish to point out that a good indicator of VIA's robustness to noise is that it works
well on a wide range of real complex datasets, spanning various numbers of lineages, population
abundances and wide ranges of feature and cell counts. In particular, a trait common to all scRNA-seq
datasets is the high level of "drop-out" or zero-inflation of counts, which in turn intensifies challenges
related to high dimensionality and unbalanced size in rare and abundant cell types (P. Qiu 2020). To
conduct a more focused study on the impact of noise, we have added the following section in
**Supplementary Note S4**, " **Noise and drop-out in scRNA-seq data** " to the Supplementary Information:

“We use the R package “seqgendiff” (Gerard, 2020) to add noise to the real RNA-seq datasets to intensify
 the attributes of noise in real data. Seqgendiff uses binomial thinning to subsample counts using the
 binomial distribution and reduce the total count expression by a specified factor. Before applying
 binomial thinning, seqgendiff randomly selects a subset of the original features (genes) so that the results
 are not dependent on the behaviour of a few features (genes).

We first show the deterioration in VIA’s inferred topology for a synthetic 4-leaf multifurcation (Fig. S17),
 when the level of noise is increased (or the signal strength is diminished) by a factor of 2 at each step.
 Note that prior to binomial thinning, only 10% of the original features are retained (at random) from the
 original input to mitigate dependency on a few select features. Since the signal in the synthetic data is
 strong, we can recover the overall multifurcating topology and delineate 3-4 of the 4 cell fates until a
 noise-factor of around 5. At higher levels of signal atrophy (Factor 7 and 8), the true structure is
 imperceptible.

To show the impact of adding noise to a real dataset (such as the endocrine dataset where the Delta
 lineage is very small, and the Epsilon lineage is easily merged with the Alpha islet cells), we compare VIA
 to other methods in terms of the inferred topology and predicted gene trends which are computed based
 on the single-cell level lineage probabilities and cell ordering. Fig. S18b-c UMAPs illustrate the level of
 added noise showing the separation between lineages becoming obscured in the noised-data
 visualization. A thinning factor = 1 was used on 25% of genes as adopted in Gerard 2020 and the
 package vignettes. VIA’s inferred trajectory remains true to the expected progression from endocrine
 progenitors, then to a forking at the Fev+ intermediate state towards different islets. More importantly,
 STREAM, Slingshot and Palantir appear to suffer most from the added noise as seen by the high
 ‘cross-talk’ between the plotted lineage trends (e.g. in the Palantir and Slingshot trends), or the detection
 of only two islets (e.g. Palantir), or in the case of STREAM the compromised topology where Alpha/Beta
 cells are plotted along the same branch, and the Epsilon/Delta are likewise combined.”

Fig.
 Impact of
 noise on
 VIA’s
 inferred
 topology
 synthetic
 leaf

multifurcation. Noise is added using

R package “seqgendiff”. 10% of the original 1000 features are sampled and retained. Then binomial thinning is
 applied where the aggregate signal is lowered to $2^{-\text{factor}}$ its original strength.

Fig. S18: Marker Gene Correlation for Predicted Lineages on Noisy Input

**Fig. S18 Gene correlation for predicted lineages on the “noised” pancreatic endocrine dataset.** 25% of quality
 filtered genes are retained, followed by factor 1 binomial thinning (a) VIA topology for noised input shows the desired
 progression of cells through 2 channels of Fev+ cells towards automatically identifying the 4 islets (red circles). (b)
 UMAP of original input (c) UMAP after applying binomial thinning blurs boundaries between the islets (d) comparison
 of inferred topology and gene trends shows that many methods merge lineages together and fail to detect 4 distinct
 islets corresponding to the Alpha, Beta, Delta and Epsilon lineages

*R1.Q2 For graph edge accuracy, which is computed based on an F1-score. I think a false negative is:*
*there exists an edge to connect to cell types that are not connected in the reference.*

**R1.A2.** We thank the reviewer for the enquiry into the F1-score pertaining to graph edges. The F1-edge
score is defined as follows: A false negative edge in the inferred model is when there is an edge in the
reference model connecting clusters/cell types/groups that is absent in the inferred trajectory. A false
positive edge in the inferred model is when an edge occurs in the inferred model that is not actually
present in the reference model. A true positive edge is one that exists in the inferred model that also exists
in the reference. These three measurements are used to compute the F1-score.

$$F1 = \frac{tp}{tp + 0.5(fp+fn)}$$

*R1.Q3 In Fig2.b, when you varied K in (KNN), the Graph edge accuracy of monocle3 is not changed. I*
*wonder whether it is true or not? Can you explain this?*

**R1.A3.** For this particular dataset, the trajectory inferred by Monocle3 did not change with respect to the
value of K in KNN. Sample outputs for Monocle3 on this cyclic dataset (**Fig. S3c**) do not change in terms
of the milestones or edges and therefore result in the same F1-score. This is driven by the invariance of
the UMAP components for this dataset, which is the input to the clustering and subsequent TI.

To elaborate further on the question of Monocle3's behaviour to changing K, we often found that
Monocle3's inferred trajectory did not necessarily reflect increased connectivity for higher K, or
conversely, diminished connectivity for lower K. For instance, in all the disconnected synthetic datasets,
Monocle3 tended to add edges between disconnected components. We found that decreasing K did not
reliably alleviate this problem. However, we rather surprisingly found that for several of the real datasets
which we know have a common precursor cell type (e.g. Pancreatic endocrine progenitors, hematopoietic
stem cells), the lineage specialized cells would be disconnected from the main body of early and
intermediate state cells, and thus unreachable from the progenitor cells. We could not reliably encourage
connectivity by simply increasing K. As such we found that the "super groups" implied by Monocle3 do
not necessarily imply truly disconnected trajectories. This suggests that perhaps the highly connected
trajectory of the MOCA 1.3 Million cells revealed by VIA offers more information about the progressive
specialization of cells, compared to the Monocle3 interpretation which offers a more disconnected and
clustered view (**Fig. 7**) .

*R1.Q4 Besides, since you compared VIA with Monocle3, PAGA, Slingshot..., what are the parameter*
 *settings of these algorithms? As the comparisons with other algorithms, you are supposed to search for*
 *many other parameters which may affect the performance apart from the number of PCs and KNN. In the*
 *other experiments, this suggestion also holds true.*

**R1.A4.** We thank the reviewer for their question regarding the choice of parameters in the benchmarked
 methods. We agree there are often several parameters left to the choice of the user, with some of these
 parameters only becoming apparent after various tests and runs, as some parameters tend to be
 ‘under-the-hood’. To address this question, we have added two tables in the Supplementary Information
 outlining the choice of parameter settings and when or why we chose a non-default setting: The new
 **Table S4** summarizes the parameters selected for each dataset that were then further investigated to
 observe impact on performance. The new **Table S5** summarizes the choice of parameter settings for each
 method.

In general, our approach was to follow the approach shown in tutorials (Jupyter NB) of other methods
 which recommend parameter settings and in some cases indicate how to deviate from default parameters.
 In general, we found KNN and/or PC to be key parameters common to all methods and therefore chose to
 focus our benchmarking on these.

Process	Type	Dimensions	Starting Point	Range of parameters varied	Remarks
Synthetic	DynToy	1000 cells x 1000 features	M1 (Milestone 1)	K-NN=20, PC=10 for composite accuracy	Each composite score comprises 5 metrics and thus we fixed the number of KNN and PCs for calculating the composite accuracy of synthetic data
Hematopoiesis	scRNA-seq	5500 cells x 14851 genes	HSC	K-NN={10,20,...,80} PC = {10,20,...,200}	Both knn and PC are varied to conduct a more comprehensive gridsearch
Hematopoiesis	scATAC-seq	2034 x 8192 TF	HSC	K-NN=20 PC = {10,20,...,200}	We consider the results for two different pre-processing pipelines and limit the parameter range to only varying the PCs
Pancreatic	scRNA-seq	2531 x 10,000 ?? genes	Endocrine Progenitor	HVG = {300,500,1000,2000,...,10000} PC = {30,40,...,80}	Here we primarily consider the effect of varying the number of HVGs
MOCA	scRNA-seq	1.3 Million Cells	E9.5 Epithelial cell	K-NN={20,30,...,50}, PC = {30} PC = {20,30,...,50}, KNN = {30}	We visually compare outputs of methods that are able to process this dataset within 3 hours
Cardiac	scRNA-seq/ scATAC-seq	695 cells+ 197 cells	Cardiac Progenitor	K-NN=10 (knn=15 for Palantir which otherwise fails at lower knn=10)	Here we primarily consider the effect of varying the number of PCs. K (KNN) is set to a low value since the cell count is small
mESC mesoderm	CyTOF	90,000 cells x 28 antibodies	Day 0 cell	All features used w/o PCA, except Slingshot which could not handle the full-feature set at high cell counts K-NN = {10,20,30,...}	Sample time annotations allow us to compute pseudotime correlations for this dataset for different K values using all 28 antibodies directly (except Slingshot which required PCA to handle the high cell count)
Cell cycle	Imaging	4000 cells x 38 morphological features	G1 cell/ G1 cluster	All features are used K-NN=20	The 38 features (without PCA) are run directly. Subsets of these features (e.g. excluding volume related features) are probed

**Table S4: Summary of key parameters considered in benchmarked datasets.** In particular, the table also
 summarizes the choice of root (i.e.starting point of the trajectory) and the rationale for which parameters are probed
 to impact TI for each dataset.

Method	General Parameter settings with emphasis on why/when parameters are changed from default	Starting point selection
VIA	 Underlying PARC settings were left at default graph pruning and Leiden resolution. We showed that good results are achievable using other clustering such as kmeans (n_clusters set to 20 as a default) which shows that the VIA method is not overly dependent or sensitive to clustering methods, as long as a sufficient resolution is captured. 	Provided as a single cell index corresponding to an early cell for real datasets; provided as a group-level label for more data-driven starting point selection for Synthetic data where Milestone group annotations are available
PAGA	 Set the number of KNN for both the PCA and diffusion map iteration The number of PCs and diffusion components in both knn-graph stages is set to the number of PCs selected rather than allowing the diffusion components to be subset to the default 10 DCs To plot gene trends we need to manually select appropriate terminal clusters and then manually select a corresponding set of clusters that make up a lineage-pathway 	A single cell index corresponding to a suitable early cell (the same starting cell is given to all methods requiring a single-cell root)
Slingshot	 KNN is not a parameter in Slingshot and hence not 'grid searched'. We first tested the method using the recommended GMM clustering but found this to be too coarse to detect the relevant lineages. The other option was using KMeans where we typically set K=15 to allow for slight overclustering without adding too much fragmentation. Slingshot's visualization of lineages is linked to the input PCs which makes it difficult to visualize results on more complex data where 2 PCs are not a good basis for visualization. 	Root manually selected as the cluster containing HSCs and the 'start' of the trajectory
Monocle3	 Monocle3 requires UMAP to be applied to the PCs before computing T1. This is recommended on their tutorial pages as part of their intended pipeline and also prompted by the tool when running it. As such, the KNN is set as the number of neighbors used in the UMAP computations and the PC value is the number of top PCs used as UMAP input. The default distance metric set by Monocle in using UMAP is "cosine distance" but this works very poorly for the datasets at hand and we therefore switched to Euclidean distance which significantly improved results 	Root branch is manually selected after getting the branching structure and selecting the most suitable milestone based on the ground truth annotations of early cells (progenitors, stem cells)
STREAM	 For STREAM, the original use case suggests to run PCA followed by one of UMAP, MLE or Spectral Embedding (SE). To ensure consistency in benchmarking and reduce the impact of additional feature reduction and subsetting incurred by a second stage of dimensionality reduction, we chose to only perform the PCA step and skip further dimensionality reduction (a more detailed explanation is given below). SE was generally too coarse for complex real data (often yielding a simple bifurcation) and MLE tends to overfragment the data such that almost each cell type (early, intermediate and late) corresponds to a "leaf branch". MLE was thus sensitive the number of components retained and quickly reached a breakpoint where overfragmentation occurred. UMAP, like SE, was often too coarse and limited the granularity. Due to the uncertainty introduced by choice of secondary Dimension reduction (with no clear winner), we choose to apply only PCA and skip the second dimensionality reduction step. This also makes for a fairer comparison to palantir, paga, Slingshot and VIA which are computed on the PCs. For the Cardiac scATAC-seq /scRNA-seq datasets, Stream would not run on the PCs (i.e. produced errors) so we used SE after PCA. We follow the tutorials provided by STREAM to incorporate a two stage approach suggested for more complex data. This allows you to refine the graph by adding more branches and nodes in a second iteration. All other parameters such as the number of initial clusters, the incremental clusters used in initializing and seeding the graph, were left as default (as per the tutorial). KNN is not a relevant parameter unless SE/UMAP/MLE are used. 	Root branch is manually selected after getting the branching structure and selecting the most suitable branch based on the ground truth annotations of early cells (progenitors, stem cells)
Palantir	 Set the number of PCs and KNN in run_diffusion_maps() and run_pca() function and override the default PC and KNN settings which otherwise subset the DCs to 10. The number of sampling points is left at the default 1500 as Setty et al., 2019 show robustness to this parameter 	A single cell index corresponding to a suitable early cell (the same starting cell is given to all methods requiring a single-cell root)
CellRank	Ran as shown on the tutorial page to reproduce the results shown in their paper for endocrine data. Skipped for other datasets as they did not have unspliced counts	Automatically determines the starting point. Since this would sometimes incur spurious starting points, we compared gene trends of the pancreatic dataset for an output which detects the correct root.

**Table S5: Summary of settings selected for benchmarked methods.** Method-specific parameter settings and
rationales for non-default choices are provided for each of the benchmarked methods.

*R1Q5. In Fig3, if you would like to compare VIA with other algorithms under different numbers of PCs*
*and KNN, a grid search of all the integers between 20-200 and 10-70 is more reasonable.*

**R1.A5.** The suggestion to make the stress-tests more comprehensive and uniform across methods and
datasets is well received. In the revised manuscript, we added a new systematic benchmarking that
ensures regular increments along the number of PCs or KNN and also included Monocle, PAGA
(whenever possible) and STREAM in all benchmarking analyses across datasets (as highlighted in our
response to Editor's comment) . In new **Table S5** we further explain which parameters are varied (i.e.
KNN, or PC, or HVGs) and why we consider a particular set of parameters for different datasets.

*R1.Q6. Also, since the clusters are detected by PARC, I am interested in whether the robustness to the*
*number of PCs is mainly because of PARC. Can you further confirm this? In the experiment of E15.5*
*murine pancreatic cells, you said that 'VIA detects small endocrine Delta lineages and Beta subtypes'.*
*Which part of the algorithm leads to this? PARC, or the lazy-teleporting MCMC?*

**R1.A6.** We thank the reviewer for this interesting question regarding the dependency of VIA's
lazy-teleporting random walks on PARC, to robustly delineate rarer but distinct cell types. We have
analyzed this further, and the reviewer is also invited to read the response to Editor (Point # 3). We added
a new section investigating this issue in the **Supplementary Note S6**, which the reader is also referred to
from the Methods section (" **VIA Algorithm** ") pertaining to the clustering step:

" *If the user prefers another clustering method or has group-labels of cell types according to apriori*
*information, VIA can easily accommodate such a substitution and the robustness of the lazy-teleporting*
*random walks to different clustering approaches is shown in **Supplementary Note S6 and Fig. S30-S32***
*for both real and synthetic data* "

As also explained in **Supplementary Note S6** :

" *A key step in VIA is to reduce the single-cell graph into a cluster-graph using the PARC's community*
*detection algorithm. We assessed the performance of VIA when substituting PARC with another clustering*
*method (in this case we chose Kmeans as it is representative of extremely simple but fast clustering) to*
*show that VIA's teleporting-random walks still traverse and uncover the expected trajectories and their*
*associated lineage-cell-fates.*

*The analysis shows two key characteristics, the first is that VIA's performance is not limited to using*
*PARC in the clustering step (**Fig. S30a**), and the second is that given the continuous nature of the*
*underlying synthetic data (**Fig. S30b**), VIA's lazy-teleporting random walks still work well even when the*
*data is not inherently highly clustered. We first tested performance on the synthetic datasets and*
*compared the composite metric scores of VIA with PARC and with Kmeans. The composite score allows*
*us to quantify accuracy of topology, pseudotime and F1-cell fate prediction for all synthetic datasets*
*across various topologies and demonstrate that VIA using Kmeans yielded competitive results (**Fig.***
***S30a**).*

*Following this, we also show that the expected gene trends and progression of cells can be recovered on*
*real data (scRNA-seq hematopoiesis and endocrine genesis). In **Fig. S31-S32** , we not only show that the*
*topology uncovered remains true to the known biological progression towards committed lineages (as*
*verified through reference to literature of the hematopoietic and endocrine genesis processes), but also*
*that the gene expression trends of known marker genes plotted versus the inferred pseudotime for each of*
*the predicted lineages, reflect the expected upregulation. We set 20 as the number of clusters, however, at*
*n=10 clusters, the megakaryocyte cells are absorbed into the larger erythroid lineage, and the dendritic*
*cells are merged together or at times even absorbed into the monocytic lineage).*

*Thus, a drawback of using Kmeans is that the choice of number of clusters needs to be sensible to capture*
*the desired resolution. In a classical clustering scenario, this might be more problematic as the choice of*

number of clusters can incur fragmentation of cell types in order to uncover a smaller population, thus
 generating 'false clusters', or merging of dissimilar cell types if the number of clusters is set to be too
 low.

To avoid the issue of setting the number of clusters, methods like PARC uncover the underlying
 granularity in a data-driven manner (without requiring a predetermined number of clusters). In PARC,
 the underlying graph pruning allows it to automatically detect rarer cell types. However, for the datasets
 considered, we found that setting K to a sensible number that captures populations of various abundances
 along the trajectory, still allows the relevant neighborhood information to be subsequently propagated in
 the random walks. These results ultimately allow the user more flexibility in their choice of how to group
 or cluster cells or potentially even use prior knowledge of cell types to guide the inferred trajectory. ”

Fig. S30: VIA robust to choice of clustering method

**Fig. S30 VIA with Kmeans clustering in lieu of PARC in the clustering step of VIA for synthetic data maintains**
 **accuracy of TI (a)** VIA with Kmeans (VIA-Kmeans) and the original VIA, i.e. VIA with PARC, (VIA) are comparable in
 terms of performance on synthetic datasets, with both approaches maintaining a competitive advantage over other
 methods, particularly for multifurcations and disconnected and connected topologies (b) Synthetic datasets plotted
 with first two principal components show the underlying data is continuous and not composed of discrete clusters.

Fig. S31: VIA using KMeans on Pancreatic Islets

**Fig. S31 VIA with Kmeans clustering in lieu of PARC for endocrine genesis data (a)** At 20 clusters, VIA
 automatically detects the 4 islets as terminal states (red circles). The topology captures the progression of lineage
 commitment arising after progressing from Ngn low to high EPs, followed by intermediate Fev+ cells. **(b)**
 unsupervised gene expression vs. pseudotime of lineage markers show the unique elevation of specific genes with
 minimal cross-talk from unrelated lineages. The two Beta subtypes are detected, with the Beta-2 subtype expressing
 Ins2 but not Ins1

Fig. S32: VIA using KMeans on scRNA-seq Hematopoiesis

**Fig. S32 VIA with Kmeans clustering in lieu of PARC for scRNA-seq Hematopoiesis (a)** At 20 clusters, VIA
 automatically detects the 6 lineages specific to the pDC, cDC, Monocyte, Megakaryocyte, B cell and Erythroid cell
 fates **(b)** unsupervised gene expression vs. pseudotime of lineage markers show the unique elevation of specific
 genes with minimal cross-talk from unrelated lineages. Notably ITGA2B (CD41) is elevated in the Megakaryocyte
 lineage and CD123 is elevated in the DC lineage representing the pDCs, and CSF1R is elevated in the cDC lineage

*R1.Q7. For Fig.7, since Slingshot is an algorithm that orders each single cell on the inferred cell lineage,*
*while VIA just infers the trajectory of cell clusters. It seems unfair to compare these two algorithms*
*considering time consumption.*

**R1. A7.** We would like to take this opportunity to clarify that although VIA begins by reducing the
single-cell graph to a cluster-level graph, VIA also ultimately provides a single-cell level pseudo-temporal
ordering and single-cell level lineage-association probability which is used to plot the lineage-pathway
and gene trends towards each individual predicted cell-fate. The projection of VIA's inferred results to a
single-cell level is included in the reported runtimes. We have now made this clearer within the main
algorithm description of the main text as follows:

“ *The single-cell level KNN Graph constructed in **Step 1** is used to project the lineage probabilities and*
*temporal ordering derived from the cluster-graph topology onto a single-cell level. Together, these four*
*steps facilitate holistic topological visualization of TI on the single-cell level (e.g. using UMAP or*
*PHATE 14,15) and critically enable data-driven downstream analyses such as recovering gene expression*
*trends and single-cell level pathways of lineages, that are essential to biological validation and discovery*
*of lineage commitment (Methods) (**Step 5 in Fig.1**).”*

We note that Slingshot's first stage of TI constructs an MST on the clusters of cells to delineate lineages
of clusters. These lineages are then used to construct principal curves, where the pseudotime of a cell is
the arc distance along a curve from the root. (Street et al. 2018). Thus, the single-cell level lineage
association probabilities and pseudotimes of VIA and Slingshot offer comparable levels of resolution.

In the main text, we typically show the graph-level trajectory as this often succinctly summarizes the
overall topology and does not require an additional step and choice of manifold embedding. However,
notable examples of the single-cell values of pseudotime ordering are shown in **Fig. 6b-c** where the 1.3
million cells of MOCA are colored by the VIA inferred single-cell pseudotime, and the single-cell
probabilistic lineage pathways for different lineages within the major groups are shown in **Fig.6c**. In the
case of MOCA we projected the results onto the PHATE embedding, but the embedding method can be
chosen by the user and the single-cell level values of VIA are independent of the type of visualization
algorithm. Note that the cytometry datasets (CyTOF of 90K cells and FACED images of 3000 cells) are
also displayed at both the graph and single-cell levels in **Fig. 7a** and **Fig. 8c,e**. Additionally we show the
single-cell temporal ordering and lineage pathways for other endocrine and hematopoietic datasets in **Fig.**
**S7, S10, S13, S14 and S19**.

*R1. Q8. VIA first represents the single-cell data in a k-nearest-neighbor (KNN) graph where each node is*
*a cluster of single cells. However, VIA cannot infer the trajectories of single cells. Since the cell fate*
*decision and changes of gene expression in single cells (especially the cells around bifurcation) are more*
*important. Can the lazy-teleporting MCMC be applied to single cells in the PARC clusters to derive single*
*cell trajectory?*

**R1.A8.** We thank the reviewer and think there are a few facets to this question. First, as we clarified in
our previous response (**R1.A7**), VIA does offer a single-cell level temporal ordering that can be used to
analyze the gene trends along pseudotime at the single-cell level. Second, the granularity of the
cluster-level graph is also easily controlled using the resolution parameter. The underlying cluster-graph
also makes VIA readily access much larger datasets where the global topological picture can be obscured
at a single-cell level. Third, since we are able to extract single-cell lineage and temporal ordering of cells
using the properties of the cluster-graph and neighborhood information of the original single-cell graph,
we can perform unsupervised prediction of key gene changes beyond up/down regulation, and even infer
more subtle changes like the slow-down in the S-phase of cell growth in the cell cycle of
FACED-image-based cytometry features; predict the synchronized upregulation of GATA1 in the
erythroid lineages, accompanied by the suppression of the GATA2 gene; recover different rates of change
of gene expression in the pre-B cell differentiation (**Supplementary Fig. S6**) which closely mirror the
reference expression levels which are annotated by sampling time (in hours).

While in theory it would be possible to simulate the lazy-teleporting random walks at single-cell level, it
would be impractical to extend the method to this level in terms of the computational cost and runtime.
Throughout our manuscript, the tests on our datasets have shown that the cluster-graph approach does not
sacrifice accuracy in terms of revealing accurate topology, cell fates and analyzing the gene/feature trends
of cells ordered at the single cell level. Moreover, global relationships are better retained at the
cluster-graph level, which is key to interpreting massive datasets like MOCA.

**Minor comments:**

*R1. Q9. The 2D embedding plots of VIA and benchmarking methods in Figure 2 seem to be colored in*
*different ways and the color legends are not displayed, which may be confusing.*

**R1.A9** Thank you for pointing this out, we have now added legends for the different outputs to guide the
reader.

*R1. Q10. The sizes in the figure texts are not unified and some of them are too small to read.*

*Fig1, caption, line84, 'can remain (orange arrows)' -> 'can remain Lazy (orange arrows)'*

*Fig2.a, M6 is missing on the first graph.*

**R1.A10.** Again, thank you for pointing this out, we have amended these typos accordingly too.

**REVIEWER #2 (Expertise: Cell cycle analysis using flow cytometry data):**

The manuscript describes the development of a trajectory inference of large single cell datasets generated
for example by scRNA-seq, imaging mass cytometry or imaging flow cytometry. There are a plethora of
algorithms designed to do exactly this task and the manuscript demonstrates advantages over a selection
of these algorithms using a range of toy and real world data.

*R2. Q1. However this is a fast moving field and new algorithms appear almost monthly which appear to*
*be capable of solving the same problems this approach is designed to be optimised for. For example how*
*does this algorithm compare with*

- • *STREAM (Chen et.al. Nat. Comms, 2019) - which compares performance with the algorithms*
*benchmarked here such as scTDA, Wishbone, TSCAN,SLICER, DPT, GPFates, Mpath, SCUBA*
*which are not benchmarked in this work.*
- • *TinGa (Todorov et. al., BioInformatics 2020) - which appears to work well for the cycle and more*
*complex geometries.*
- • *Tempora (Tran et.al. PLOS COMPUTATIONAL BIOLOGY 2020)*

**R2. A1.** We thank the reviewer for their suggestion to consider other methods that have also been
developed with the aim of achieving complex trajectory inference. We selected the methods we
benchmark based on strong performance in a recent benchmarking analysis (Saelens et al. 2019) as well
as certain prerequisites shown to be satisfied within their papers, i.e. the ability to automatically predict
cell fates and lineages on non-tree topologies, scale to large number of cells and even handle various omic
data types (i.e. Palantir, Slingshot, Monocle3). We note that although PAGA does not provide automated
cell fate prediction, its scalability and flexibility with regards to topology and single-cell modalities, and
the ability to integrate DPT to extract single cell pseudotime, make it largely relevant to our analysis.
Here we outline our rationale for choosing whether or not to include STREAM, TinGa and Tempora:

STREAM (Chen et al., 2019): We have now included STREAM in our benchmarking analysis for all
datasets (real and synthetic) across all stress-tests. STREAM offers pseudo temporal ordering and cell fate
lineage prediction on real datasets and can be included in most of the comparative analyses within our
paper. We note that the methods Chen et al., chose to benchmark in their paper all performed significantly
worse in the Saelens et al. 2019 benchmarking study, than the methods we already included in our paper.
It focuses on tree-like trajectories and also lacks scalability (taking 4 hours on high-dimensional data of
500 features and 2000 cells, and over 6 hours on high cell count of 1 Million MOCA cells of 30 PCs).

TinGa (Todorov et al, 2020): While TinGa is an interesting new method, it does not predict cell fates or
lineage pathways which we view as an essential part of real-data analysis to facilitate unsupervised
analysis of lineage committed cells, the differentiation pathways and the gene expression along their
respective lineages. Without this information, TinGa would require manual setting of cell fates and the
associated lineage pathways in order to be included in the benchmarking conducted on the real datasets in
our paper. We also noticed that Todorov et al.'s testing of real data is generally limited to smaller and
simpler scRNA-seq datasets. As such, STREAM appears to be better suited to the benchmarking
framework used in this paper.

Tempora (Tran et al., 2020): This is also an interesting tool but is tailored to transcriptomic data which
(preferably) have temporal annotations that can be used to guide the trajectory. Tempora is also designed
to incorporate Gene Ontology terms to create a pathway enrichment profile which removes redundant
pathways. Tran et al., show that without the use of GO terms - which would be the case for non
transcriptomic data - the accuracy of Tempora (as Tran et al. show in their paper) falls below that of
methods they benchmarked.

Additionally, Tempora does not automatically detect cell fates and their lineage pathways or offer
single-cell level ordering of cells, all required for the unsupervised plotting of gene trends along detected
lineages. This again makes it difficult to effectively benchmark, particularly for real data.

We do acknowledge that in the case of time series data, it is useful to incorporate the sampling time labels
and is a feature we would consider adding to VIA in future releases. Such a feature would resemble
FlowMAP which uses the adjacency of temporal annotations to determine the graph-edge connectivity.

We therefore consider the set of chosen methods for benchmarking - now including STREAM - to be a
fair representation of ‘state-of-the-art’ methods. We show below (**R3.Q1**) the performance of STREAM
on the synthetic datasets and refer to our other dataset specific figures in the main text and gene- trend
plots (**Supplementary Fig. S9, S11, S15**) for comparison on real data. We have also added a note in the
revised main text and method section highlighting the general rule-of-thumb for choosing methods for
benchmarking:

“...lazy-teleporting random walks allow VIA to consistently identify pertinent lineages that remain
elusive or even lost in other top-performing and popular TI algorithms we benchmark (Saelens et al.,
2019), which are chosen for comparative analysis conditional on meeting several of the following
criteria: automated lineage path and cell fate prediction, recovery of complex topologies not limited to
trees, scalability and generalizability to multiple single-cell-modalities.”

In the Methods section:

“The methods were chosen based on their superior performance in a recent large-scale benchmarking
study⁴, including a select few recent methods claiming to supersede those in the study. Specifically, recent
and popular methods exhibiting reasonable scalability, and automated cell fate prediction in
multi-lineage trajectories, not limited to tree-topologies, were favoured as candidates for benchmarking
(See **Supplementary Table S1** for the key characteristics of methods). Performance stress-tests in terms of
lineage detection of each biological dataset, automated gene trend prediction along lineages, and
pseudotime correlation were conducted over a range of key input parameters ... Methods that focus
exclusively on a single data modality or on topology without predicting cell fates and their lineage
pathways (e.g. Tinga, Tempora) were generally not included in the benchmarking as they would require
manual selection of cell fates and differentiation pathways.”

**R2.Q2.** An example is given of the analysis of the cell cycle progress of cells using Imaging Flow
Cytometry. However this type of analysis has been carried out previously [Blasi *et.al.* 2016, Eulenburg
2017 - Nat Comms]. The trajectory is not very complex so a simple diffusion map approach can work.

**R2.A2.** We thank the reviewer for drawing our attention to two very interesting papers using imaging
flow cytometry to analyse the cell cycle process. Blasi et al. showcased the ability of trained machine
learning algorithms to successfully predict cell stages based on extracted biophysical features. Similarly,
Eulenburg et al. also focused on a CNN trained by labeled cell cycle stages and then subsequently used
the last layer of the CNN as the input to a t-SNE visualization of the cell cycle progression trend. While
they represent valuable findings, we believe that VIA offers a different use case where the labels of cells
are not known in advance for graph construction, temporal ordering and the subsequent feature trend
analysis. The inclusion of label-free biophysical phenotypes that were absent in the above two studies (i.e.
single-cell mass-density and its subcellular spatial distribution), combined with the application of VIA on
FACED imaging flow cytometry demonstrates the new significance of combining label-free single-cell
biophysical profiles with advanced TI methods (VIA in this case) .

Whilst the trajectory of the FACED image based cell cycle is not complex, we have now added the new
**Fig. S23** (also below) to illustrate that Palantir and ‘PAGA+DPT’, two methods which use Diffusion
Maps do not provide a clear interpretation of the linear progression shown by the cells. For instance,
PAGA forms S-shaped loops and edges which distort inference of the cell-cycle-related morphology.
Palantir is highly sensitive to the choice of root cell as the diffusion map representation disperses G1 cells
along the entire trajectory which means that Palantir often finds multiple ‘terminal’ states which are in
fact G1 stages. Also, we note that STREAM places S-phase cells in a separate branch indicating a
bifurcating structure. Moreover, VIA is the only method which shows the slow-down of the cell growth
during S-phase.

Fig. S23: FACED - other TI method outputs

**Fig. S23 FACED biophysical profiles of cell-cycles analyzed by other methods.** The topologies and the temporal
 trend of volume inferred by other methods is shown for MCF7 and MDA-MB231. Note that for PAGA and STREAM
 we manually select the branches or clusters used in the trend plots. In STREAM the S cells are placed in a separate
 branch in a bifurcating structure. PAGA structure has an “S-shape” which we do not follow when selecting suitable
 clusters from G1 to G2 state. Many methods such as Slingshot and Palantir show multiple lineages ending at different
 stages along the cell cycle even though the root cell is selected as a G1 stage at the ‘tail end’ of the data
 representation.

*R2.Q3. The authors also use an ergodic approach (Kafri et.al,2013 Nature) to infer details about certain*
 *cell characteristics, however as Kafri discussed, care must be take to not use any predictor measure*
 *corrected to the response variable i.e. if you are inferring dry mass, volume then no correlated variables*
 *(e.g. cell radius, area etc) should be used in the pseudo time alignment.*

**R2.A3.** We thank the reviewer for this query. Taking the reviewer’s suggestion, we probed the features by
 removing volume and all features highly correlated with volume. Only features satisfying an absolute
 correlation with volume below 80% (excluding features such as cell dry mass, area, volume) were used to
 uncover the same size-related trends. The new results are presented in **Fig. S25-S26** .

Running VIA without the cell volume (and volume-correlated) features, we see that VIA still correctly
 uncover a clear cell-cycle progression and only identifies G2 stage cells as cell fates. It also recovers the
 slow down in growth during the S-phase. The results are consistent across the two different cell lines (i.e.
 the MCF7 and MB231 datasets). We also compare the results to PAGA and Palantir (which use diffusion
 components) to show that relying on the diffusion component space to infer the trajectory does not
 necessarily offer satisfactory results. For instance, Palantir identifies multiple intermediate cells as final
 cell fates even though we have carefully provided Palantir with a starting cell at a tail end of its data
 representation. The graph topology of PAGA, on the other hand, suffers from spurious edges that could
 confound intuitive interpretation and analysis of TI (e.g. **Fig. S23, S25-S26**).

We would also like to clarify that in our original analysis, the volume (like any other feature) serves to
 guide the temporal ordering of the cells, and is not used to predict the actual values of other features. The
 inferred temporal ordering is then used to plot feature expression, thereby allowing us to visualize the
 changes in expression according to the inferred chronology. We find this to be the standard approach
 taken by other methods on other omics datasets where marker-gene trends are plotted even though the
 same gene (or highly correlated ones) were used in the TI.

**Fig. S25 Cell-cycle TI predicted without cell volume (and volume-correlated) features (MDA-MB231 cells).**
 *Volume and features with $|Corr(x, volume)| > 0.8$ are removed from the input to trajectory inference. We then plot*
 *volume along the inferred cell ordering to see whether we detect the correct trend, including the slow-down in*
 *S-phase. Note that for PAGA, the clusters from a root cell to a final “G2 cluster” are manually provided.*

Fig. S26: MCF7 cell cycle TI w/o size related features

**Fig. S26 Cell-cycle TI predicted without cell volume (and volume-correlated)**
features (MCF7 cells). Volume
 and features with an absolute correlation with volume above 80%, i.e. $|Corr(feature, volume)| > 0.8$, are removed from
 the input to TI. We then plot volume to see if the correct trend along the inferred cell ordering is observed, including a
 slow-down in S-phase. Note that for PAGA, the clusters from a root cell to a final “G2 cluster” are manually provided.

**R2.Q4.** I believe the manuscript is probably better suited to a method type journal as it appears to be an
 addition to an already large set of algorithms to do this job and also details of some of the examples need
 to be further elucidated as discussed above.

**R2.A4.** While we acknowledge that trajectory inference is a rapidly developing field, pushing the limits
 of current methods in terms of the scalability along feature and sample size, and the generalizability to
 various topologies has neither been obvious nor straightforward (as discussed in our introduction and
 evident in the results presented in this work). In this manuscript, we present VIA as a new TI method,
 using the new strategy of lazy teleporting random walks, to demonstrate its power in many challenging
 single-cell TI applications (particularly discovering elusive lineages and rare cell fates) across different
 omics datasets, including the emergent biophysical (imaging) cytometry .

Furthermore, many TI methods tackle either topology (e.g. PAGA, Tempora, TinGa), or the single-cell
 level ordering and lineage trends without a topological abstraction (e.g. Palantir, Slingshot). In contrast,
 VIA provides an analysis that can integrate the topology with single-cell level ordering and lineage
 prediction to allow biologists to perform a more multifaceted analysis. Another key practical
 consideration is that the speed of VIA allows users to experiment with different parameters and
 pre-processing methods as multiple runs are inexpensive.

Overall, VIA represents an important technical advancement as well as a unique facilitator for new
 discovery by allowing biologists to design and explore multi-modal, large-scale experiments. We note that
 the broad interest in these key strengths is evidenced by VIA’s altmetric score of 46 on BiorXiv (top 5%
 of research) . We believe the advances presented here are sufficiently impactful in the ever-growing
 single-cell omics community.

**REVIEWER #3 (Expertise: trajectory inference using single cell data):**

*R3.Q1. This paper presents the VIA method for inferring trajectories in single-cell data, regardless of the*
*–omics type. It relies on one metric, the F1-score (that assesses how accurately cell fates were identified*
*by each method), to assess the accuracy of the VIA method and compare it to 5 state-of-the-art methods.*
*This F1-score seems too weak of a metric to me, as it doesn't capture the structure of the returned*
*trajectory. I can think of different trajectories (successively bifurcating, trifurcating, or even*
*disconnected), that could have exactly the same cell fates, and thus return the same F1-score. I would*
*thus recommend adding a second metric, next to the F1-score, that would capture the topology of the*
*different trajectories. Moreover, the F1-score metric that was chosen by the authors doesn't allow them to*
*compare all the tools that they included in the study (as PAGA doesn't automatically detect terminal*
*states). If the paper chose to use only one metric to compare all methods, this metric should at least allow*
*to compare all methods. In my opinion, the paper would benefit from adding at least one more metric to*
*compare the methods on.*

**R3.A1.** We thank the reviewer for the suggestion to add another metric and more directly evaluate
accuracy in terms of the topology of the inferred trajectory. We agree that this would provide a more
holistic platform to benchmark methods, and also allow for methods which do not predict lineages, like
PAGA, to be evaluated. In the revised manuscript, we have therefore revamped the benchmarking
analysis of both synthetic and real data in the following ways:

**Synthetic datasets:** Leveraging the availability of reference models of the topologies and sampling
499 times given by the synthetic datasets, we took a comprehensive quantitative approach using a new set
of metrics to systematically evaluate the performance of each benchmarked method.

**Real biological datasets:** We deepened the gene-trend analysis along the inferred lineages to offer
both a quantitative and qualitative comparison across all relevant lineages for each method across 3
datasets which have multiple cell fates of varying population sizes, with known marker genes.

For the synthetic data we 1) defined a new set of metrics (detailed below) that assess multiple layers of
the inferred trajectory, including the topology, pseudotime and predicted cell fates and 2) added more
synthetic datasets to have two of each type of trajectory (multifurcating, cyclic, connected and
disconnected). (**See the new Fig. 2** for an overall summary and **new Fig. S2-5** for a breakdown of the
composite metric's components for each topology).

We have also added the description of these new metrics in the main text accompanying **Fig. 2** , and in the
method section. Here below is summary in main text:

*“The availability of a reference truth model for the synthetics datasets allows us to quantify TI accuracy*
*using a composite metric which assesses multiple layers of the inferred trajectory including topology,*
*pseudotime and lineage prediction. The metric measures “local” graph similarity between the inferred*
*and reference graphs using the Graph Edit Distance (GED), an F1-Branch score (which labels branches*
*as True or False Positives, or the lack thereof as a False Negative), and the “global” graph similarity is*
*computed using the Ipsen-Mikhailov measure (**Methods**). The breakdown of the composite score is shown*
*for each dataset in **Fig. S2-S5** .*

*The differences in accuracy between VIA and other methods is most notable in complex topologies,*
*particularly those with disconnected components comprising various connected topologies.”*

Further details of the composite accuracy score as well as the figure showing the individual components
 of the score for each topology are now available in the new **Supplementary Note S3** :

“We use 9 synthetic datasets to benchmark the performance of VIA and other methods on a variety of
 trajectories and topologies (see Fig. 2) . The 9 synthetic datasets are generated using DynToy and contain
 1000-3000 cells across 1000 features. They span (bi/multi) furcations, cyclic, connected (which are
 combinations of cyclic and furcating) and disconnected (which are combinations of all the
 aforementioned). The composite accuracy metric assesses multiple layers of the inferred trajectory, taking
 into account the topological similarity between the reference model and the inferred topology, the
 correlation between the real and ‘pseudo’ times, as the prediction accuracy of the terminal cell fates
 (lineages). For certain metrics, the absolute measurements of similarities or differences are converted
 into a percentage scale such that 100% corresponds to the highest score. The composite metric is the
 arithmetic mean of the 5 scaled metrics.

**Ipsen-Mikhailov (IM)** : IM distance is used to measure the similarity of global graph topology. The IM
 ranges from 0 to 1 and equals the difference in spectral densities of two graphs. $IM(G_1, G_2) = 0$ implies
 that the graphs are isospectral. Unlike the Graph Edit Distance and the F1-branch score, where only the
 structure of each link’s immediate neighbourhood contributes to the distance value, the IM considers the
 structure of the whole topology. We use $1 - IM$ for the composite accuracy score.

$$537 \quad IM(G_{TI}, G_{REF}) = \sqrt{\int_0^{\infty} [\rho_{REF}(\omega) - \rho_{TI}(\omega)]^2 d\omega} \quad [19]$$

$$538 \quad \rho(\omega) = \sum_{k=1}^{N-1} \frac{\gamma}{(\omega - \omega_k)^2 + \gamma^2} \quad [20]$$

distributions,

$\rho(\omega)$ is the graph spectral density, as a sum of Lorentz
 are the vibrational frequencies
 ω_k

given by $\sqrt{\lambda_i}$ where λ_i are the eigenvalues of the Graph Laplacian. γ is the half width at half maximum.

**Graph Edit Distance (GED)** : is defined as the cost of converting G_{TI} to G_{REF} with the least possible
 number of operations. Each operation has a cost of one and includes insertion/deletion of edges and
 nodes. Oftentimes graph edit distances like the (e.g. GED, Hamming distance) are normalized by the
 entire set of possible edges $N(N - 1)$, but this creates an inflated sense of accuracy in the case of very
 sparse graphs like the reference topologies of the synthetic datasets. We therefore normalize by the sum of
 nodes and edges in G_{REF} . 1- ‘Normalized’ GED is used in the composite accuracy measure.

**F1-Branch score**: is applied to the local branch accuracy and defined as follows . A False Negative edge
 in the inferred model is when there is an edge in the reference model connecting clusters/cell types/groups
 that is absent in the inferred trajectory. A False Positive edge in the inferred model is when an edge
 occurs in the inferred model that is not actually present in the reference model. A True Positive Edge is
 one that exists in the inferred model that also exists in the reference. These three measurements are used
 to compute the F1-score.

$$553 \quad F1 = \frac{t \cdot p}{tp + 0.5(fp + fn)} \quad [21]$$

**Temporal Correlation**: Given the sampling times of the synthetic data, we use the Pearson Correlation
 coefficient as a measure of how closely the inferred pseudotime follows the true sampling times. σ_X is the
 standard deviation of X , and μ_X is the mean.

$$\rho_{x,y} = \frac{E[(X-\mu_x)-(Y-\mu_y)]}{\sigma_x\sigma_y} \quad [22]$$

[revised manuscript text omitted]

For Slingshot, Palantir and VIA, the comparison is fairly straightforward as these methods offer
single-cell level temporal ordering as well as lineage likelihoods which can be used to infer the trends of
genes. PAGA however requires manual selection and ordering of a pathway from root to desired cell fate
before plotting the gene trend in relation to pseudotime by using a sliding window approach. For
STREAM we again selected only the desired branching pathway for a given lineage to minimize potential
cross-talk. The lack of single-cell level differentiation pathways in STREAM (where we could only select
pathways at the branch level), lowers the quality of the plotted trends. We then applied GAMs in order to
smooth the gene trends so that the computed correlations could be compared to the other methods.
Monocle3 is more challenging as its gene-pseudotime function does not consider lineage pathways but
rather orders all cells along the inferred pseudotime before fitting a curve. This oftentimes resulted in the
curve fitting being so noisy it overwhelms the original data.

Here, we briefly highlight key features in the gene trend comparison for each of the three real datasets:

**scRNA-seq hematopoiesis (Fig. S9):** The dataset contains two types of dendritic cells (the classical and
the plasmacytoid DCs). We point out however, that for most methods only a single lineage is upregulated
for the Dendritic marker IRF8. Similarly, the CD41 (ITGA2B) is often upregulated by the erythroid
lineage rather than by a distinct lineage exclusive to the megakaryocytes (e.g. Slingshot's 'blue' curve is
the only lineage showing both CD41 and GATA1). In PAGA we note that the pseudotime scale is
distorted due to the weak linkage to the CLP (B cells) lineage and the gene plots are very noisy. Monocle3
also separates the CLP lineage away from its progenitors which means it cannot be reached from the HSC
state. For other values of KNN/PCs, Monocle3 often even splits the erythroid lineage as a disconnected
group from the progenitors. STREAM sometimes places the same cell type along multiple distinct
branches which results in the additional up/down regulations seen in the predicted gene curves.

**scATAC-seq hematopoiesis (Fig. S11):** Monocle3 splits the monocyte and CLP lineage away from the
intermediate states, making them undetectable from the HSCs. For Slingshot, we find that the F1-cell fate
scores are generally quite poor, not because it fails to detect the relevant lineages, but because it detects
several intermediate stages as terminal cell fates. However, when plotting the gene trends we select only
the 4 that most distinctly capture the relevant cell fate lineages. Although the marker genes are
significantly upregulated for the selected lineages, we see that often the correlation scores are low because
the lineage pathways include cells that are irrelevant to the terminal cell fate and cause the expression
levels to drop at later stages along the branch. The visualization of Slingshot's topology (shown in the
rightmost column of **Fig. S11**) when using more than a few PCs is also difficult to interpret because the
principal curves are made on the PCA input, but cannot be linked to a t-SNE/UMAP visualization. It thus
resorts to plotting the lineage curves on the first two PCs. The coarse branching structure of STREAM
(even though the full pipeline for complex datasets is followed) again means that cells that are not very
relevant to a certain lineage are included in the branches, thus contributing to that cell fate and perturbing
the gene trends even when using the same cubic spline function as applied in VIA. As seen in the
branching structure of the VIA graph, the granularity of the VIA's branching structure allows for the
computed lineage-probabilities to more accurately reflect the likelihood of cells leading to a particular cell
fate, and consequently refine the plotted gene trends.

**scRNA-seq endocrine genesis (Fig. S15):** VIA, Slingshot and STREAM perform well on the endocrine
dataset, consistently identifying the 4 pancreatic islets. The gene trends however are superior for
Slingshot and VIA where lineages exhibit less intermingling along a determined single-cell differentiation
pathway, resulting in clearer gene expression of the lineage specific markers. VIA can distinguish
between the two Beta cell types (based on the difference in Ins1 and Ins2) even though the underlying
number of clusters is similar (PARC automatically identifies ~15 clusters, and Kmeans used by Slingshot
is set to 15).

The correlation coefficients for cell ordering along lineage marker expression are summarized in the new
subfigures in **Fig.3-4:**

Fig. S9: scRNAseq Hematopoiesis Gene Correlation for Predicted Lineages

**Fig. S9 Comparison of marker gene correlation computed by different TI methods (for predicted lineages in**
 **scRNA-seq hematopoiesis).** Automated gene expression plots as a function of pseudotime for the lineages
 associated with 6 specialized cell types. For PAGA and STREAM we manually select the order of clusters/branches
 towards a terminal state. PAGA uses a sliding window approach to compute the expressions based on the selected
 clusters, whilst for STREAM we apply GAMs to plot the expression. For Slingshot's topology, the black lineage curves
 cannot be aligned with the t-SNE embedding, but we show the colored temporal ordering of cells on the t-SNE for
 ease of comparison to other methods.

Fig. S11: scATACseq Hematopoiesis Gene Correlation for Predicted Lineages

**Fig. S11 Comparison of marker gene correlation computed by different TI methods (for predicted lineages in**
**scATAC-seq hematopoiesis).** Automated gene expression plots as a function of pseudotime for the lineages
associated with CLP, MEP, Monocyte and DC committed cells. For PAGA and STREAM we manually select the order
of clusters/branches towards a terminal state. PAGA uses a sliding window approach to compute the expressions
based on the selected clusters, whilst for STREAM we apply GAMs to plot the expressions. Monocle3 did not plot
fitting curves on this dataset.

Fig. S15: Marker Gene Correlation for Predicted Pancreatic Endocrine Lineages

Fig. S15 Comparison of marker gene correlation computed by different T1 methods (for predicted endocrine
 lineages). Automated gene expression plots as a function of pseudotime for each of the 4 pancreatic islets: Alpha
 (Gcg), Beta (Ins1 and Ins2), Delta (Sst) and Epsilon (Ghrl). For PAGA and STREAM we manually select the order of
 clusters/branches towards a terminal state. PAGA uses a sliding window approach to compute the expressions based
 on the selected clusters, whilst for STREAM we apply GAMs to plot the expressions. The dark curves on the Monocle
 plots are very noisy and overwhelm the original data points.

*R3.Q2. Small comment related to the previous one: in figure 2a), the authors state that the F1-score can't*
*be computed for PAGA as this method doesn't automatically return cell fates. However, an F1-score was*
*then computed for PAGA for the cyclic dataset (presented figure 2b), and disconnected dataset (presented*
*in figure 2c). Please clarify in the text, in the methods, or in the description of figure 2 how and why the*
*F1-score was computed for some datasets but not for others. The fact that the F1-scores presented on the*
*right of figure 2 correspond to a different set of methods for each dataset is confusing.*

**R3.A2.** We apologize for the confusion and we have changed the figure such that it is now more easily
interpretable. In the previous version of the figure, the F1-scores for the cyclic and disconnected
trajectories were F1-branch (graph-edge) scores, not F1-cell fates scores - and quantified the graph edge
topology similarity between the reference and inferred topologies. We were therefore able to include
PAGA for this particular graph-similarity accuracy measure.

*R3.Q3. It is unclear to me why, in each figure where VIA is compared to other methods, it is not being*
*compared to all the other methods. For example, why did the authors choose not to include the results of*
*PAGA in figure 4? Of Monocle3 in figure 7?*

**R3.A3.** As detailed in our response above (**R3.A2**), we have extended our comparisons for each dataset to
include all methods (including now PAGA for gene trends in **Fig. 4** and Monocle3 in **Fig. 7**). In the case
of PAGA, we generally do not perform extensive stress-testing across PCs/KNNs in most datasets as
PAGA cannot predict cell fates. We have however included PAGA in our comparison of gene-correlation
comparisons where we were able to manually select the pathway from root to a manually selected
appropriate cell fate cluster. PAGA is also part of the synthetic dataset analysis where the composite score
for PAGA is calculated by excluding the F1-cell-fate score and including the other 4 metrics. For the
MOCA dataset in **Fig. 6** , we include PAGA in the comparisons because we do not quantitatively assess
the performance as there is no clear reference truth. Therefore we can visually compare and contrast the
topologies inferred by VIA, PAGA and Monocle3 (based on the UMAP input) for various KNN and PCs
as these approaches can be reconfigured and run in a reasonable timeframe. Since there is no clear
reference topology and we need to rely on the sampling times and annotated cell types to understand this
dataset, we have provided an extensive discussion on the inferred branching structure in both the main
text as well as in **Supplementary Note S3** .

*R3.Q4. The paragraph starting at line 378 should, in my opinion, be rephrased. Historically, the first*
*trajectory inference (TI) tools were designed to be applied on cytometry data (Wanderlust, Monocle). This*
*type of data contains a much better ratio of cells over features than scRNA-seq data, and suffers less from*
*sparsity, which typically makes it easier to find trajectories in this type of data. However, the way the*
*paragraph is written now, the authors seem to suggest that most TI methods can only handle*
*transcriptomic data, and that handling cytometry data would be a new advantage of VIA. I would thus*
*advise to rephrase this paragraph to avoid misleading the reader.*

**R3.A4.** We thank the Reviewer's suggestion for further articulating the argument/rationale of the
cytometry application of VIA. Our intention is to convey that many methods are frequently only tested on
a single experimental modality, even if they can in theory be extended to other types of datasets. We

merely wish to point out that we test VIA for practical compatibility to a diverse range of omic datasets.
We have revised this paragraph accordingly as follows,

*“ Broad applicability of TI beyond transcriptomic analysis is increasingly critical, but existing methods*
*have limitations contending with the disparity in the data structure (e.g. sparsity and dimensionality)*
*across a variety of single-cell data types. We have thus far shown that VIA can be used to successfully*
*interrogate scATACseq, scRNAseq and their integrated data, and now proceed to further investigate*
*whether VIA can cope with the significant drop in data dimensionality (10-100), as often presented in*
*flow/mass cytometry data, and still delineate continuous biological processes.”*

*R3.Q5. The authors emphasize the fact that VIA doesn't require any user input parameters, compared to*
*other methods. Only when reading the methods section do we discover that VIA needs the user to define a*
*starting point to run. This should be stated earlier in the paper in my opinion (around line 25 for*
*instance), as this is an information that any potential user of the method will be interested in. Side*
*comment: as PARC allows a fast visualisation of the data, I'm wondering if allowing the user to choose a*
*starting point in this representation of the data would be a nice way to simplify the choice of a starting*
*point for the user.*

**R3.A5.** We have revised the text in the Algorithm section of the main text to indicate that VIA requires
the user to select the root node cell or cluster:

*“The root (starting point) is designated by the user either as a single-cell index or using group or cluster*
*level labels.”*

We observe that the benchmarked TI methods require either a single cell to be initialized as the root cell
(Palantir), or selected by the user as a cluster/milestone after an initial topology is constructed (Monocle3,
PAGA, STREAM). Since the initialization of a root by group-level or single-cell annotation depends on
the information available, we allow the user to choose either of the two approaches and have added the
following explanation in Methods:

*“The root cell is initialized by the user in one of two ways: If for instance there are some cell*
*type/group/cluster level labels available in advance, the desired starting group can be indicated to VIA,*
*which will then automatically select a cluster in its cluster-graph that contains a majority of this*
*particular cell type/group classification. In the case of many clusters satisfying this criteria, VIA selects*
*the cluster in the graph that has connectivity metrics indicative of a root (leaf) node (such as low*
*betweenness and low centrality). The user can also choose to provide a specific single cell as the root*
*node. In case the user wishes to select the root based on the VIA graph, one would save the*
*VIA-cluster-graph labels and use them to guide selection of the root as described in the first approach.”*

*R3.Q6. The authors highlight the fact that the VIA method's computational advantage allows it to process*
*larger amounts of dimensions, avoiding information loss due to dimensionality reduction. However, they*
*test VIA on all original genes only in one dataset (presented in figure 4), and apply VIA on PCs for the*
*other datasets. It would be interesting to see how the method performs on the original dimensions and*
*how it compares to other methods on the synthetic datasets, and in figure 3.*

**R3.A6.** We thank the reviewer for enquiring about our approach to ‘high dimensional’ feature inputs. We
first explain the approach taken in our paper for benchmarking and general performance analysis, and
then introduce the new analysis we have done to test VIA on gene-count features without any PCA (or
other dimensionality reduction). We also invite the reviewer to read our **response (point 2) to the Editor**
above where we commented on this substantial addition to the paper.

**Discussion and context of original analysis:**

In our original analysis of various datasets, we showed that VIA handles a broad range of input feature
dimensions that are used to characterize various omic datasets. Imaging and mass cytometry features tend
to remain within 10-100 features (antibodies or morphological features) which VIA parses without any
dimensionality reduction. On the other hand, scRNA-seq produces 5000-15000 genes even for small
sample sizes of hundreds or a few thousand cells and it is common to use a dimensionality reduction such
as PCA to enhance the signal in the data and avoid the curse of dimensionality, before running further
unsupervised computational analysis. Since all the benchmarked methods require PCA, and afterwards
often a second stage of further dimensionality reduction (e.g. UMAP, MLLE, diffusion maps) before
performing TI, we found that including PCA in the pre-processing pipeline allowed more consistent
comparison across all methods.

Several TI methods, including top-performing ones like Slingshot, STREAM and Monocle3 require or
highly recommend further reducing the dimensionality using a second method such as UMAP, TSNE or
the like. Other methods like Palantir and PAGA which can receive raw features (e.g. for cytometry data)
or a higher number of PCs (in the case of gene data) involve steps under the hood that subset the number
of diffusion components (default of 10 eigenvectors and an even lower number of diffusion components
in PAGA and Palantir) retained for further TI analysis. **See the new Supplementary Note S5 and Fig.**
**S27** for more details on the dimensionality steps used by the different methods. We found that forcing
other methods to compute directly on the genes by overriding dimensionality reduction and subsetting of
components resulted in distorted topologies (**Fig. S27a**).

**New analysis on full feature inputs:**

However, to stretch the capabilities of VIA, we now show that it can sensibly process both real and
synthetic data using several thousands of genes without any form of dimensionality reduction. We used
the synthetic ~1000 ‘cell’ datasets spanning various topologies to observe how faithfully VIA uncovers
topology, pseudotime and cell fate prediction when using all 1000 features. These features were provided
to VIA without any dimensionality reduction and yielded results very consistent with the underlying
‘ground truth model’ Supplementary Fig. S27 (distinguishes between “VIA-Full” and VIA-PCA, while
all other methods are run using PCA first). We refrain from a full benchmarking against other methods as
they are not designed to handle such high dimensionality. For instance, Slingshot and STREAM can take
several hours on a few thousand cells when the dimensionality exceeds 100 and Monocle3 will not run
without a UMAP embedding. PAGA and Palantir depend on using only a small subset of eigenvectors

and diffusion components, without which they produce unreasonable answers (e.g. PAGA results in a
 fully connected cluster graph as seen below in **Fig. S27** and Palantir overly fragments the data such that
 the topology cannot be interpreted visually). We therefore can only reasonably compute the composite
 accuracy metric for VIA on the full feature sets, and show the TI outputs of PAGA and Palantir when we
 override any ‘under-the-hood’ subsetting of features to show that a full comparison would not be feasible.

Fig. S27: Performance on all features w/o PCA on synthetic data

**Fig. S27 TI performance comparison with full feature input using synthetic data (no PCA)** (a) Sample outputs of
 VIA topology when using all 1000 features without PCA compared to PAGA and Palantir. Although the runtimes for
 PAGA and Palantir are reasonable, their outputs are distorted due to overriding the PCA and subsetting of diffusion
 components. (b) Composite accuracy scores of VIA when using the full-feature input (VIA-Full), compared to VIA
 using PCA (VIA-PCA), VIA using Kmeans (and PCA) and all other TI methods which use PCA. We see that VIA can
 achieve good accuracy even without the PCA step for many of the synthetic datasets.

**R3.Q7.** Comment related to the previous one: it is surprising to see that VIA performs optimally when
applied on > 6000 HVGs in figure 4, while Slingshot performs best on less than 2000 HVGs (which is
more expected). This might be due to the fact that the F1-score captures only the accuracy of the end
states recovered by the different methods, and doesn't represent differences in topology. It would be
interesting to see whether the same increase in accuracy is observed when using more HVGs in the
synthetic data and in figure 3.

**R3.A7.** We first wish to clarify that the table (in the original **Fig. 4**, now **Fig. S1c**) indicates the number
of HVGs used to compute the PCs before using a range of these PCs as inputs to the different algorithms.
In the main text and the purpose of method comparisons, we pre-processed these datasets by following
the same protocols as their authors (**See Methods and Supplementary Tables S4-S5**). As mentioned
above in **R3.Q6** , other methods do not lend themselves to considering 1000s of features without some
form of dimensionality reduction, and thus we limit the assessment of high-feature-inputs on real data to
only VIA.

For the purpose of evaluating the topological consistency and cell fate prediction when the input is a
range of HVGs without any PCA or other dimension reduction, we examine two real datasets
(scRNA-seq endocrine genesis and hematopoiesis) chosen because they are multi-lineage and have cell
populations of varying sizes (See new **Fig. S28-S29**). When using the genes as a direct input (after basic
filtering and log-normalization), we consider the choice of number of HVGs provided directly to the
algorithm as a parameter that could influence the outcome. For both datasets, we visualize the stability in
topology across a range of HVGs by showcasing the topological outputs side-by-side. The nodes are
colored by pseudotime, with the relevant lineages colored according to the legend.

**scRNA-seq hematopoiesis (Fig. S28):** In general, the hematopoietic dataset yields high-quality results
when using 500-5000 genes (**Fig. S28d**), with three main branches appearing: the CLP (B cells) branch,
erythroid/megakaryocytic branch and monocytic/dendritic branch. Within these main branches, the DCs
are typically separated (as shown by the colored branches) from the monocytes (evidenced by the unique
upregulation of IRF8 in the DC lineage, and not by the monocyte lineage), and the erythroids from the
megakaryocytes (the upregulation of CD41 in the lineage associated with megakaryocytes compared to
the erythroid lineage). As the number of HVGs exceeds 5000, the structure deteriorates. This is expected
given the number of features would begin to significantly exceed the number of datapoints (cells).

**scRNA-seq murine endocrine genesis (Fig. S29):** Surprisingly, even though the number of cells is only
a few thousand, the pancreatic dataset tolerates using up to 10,000 HVGs without sacrificing topology or
cell fate prediction. The gene trends are plotted for the automatically predicted 4 lineages (for the 1000
HVG case), and correspond to the 4 pancreatic islets. As shown by the topological outputs, the sustained
cell fate identification of the islets is also accompanied by a topology that is biologically consistent. The
Ngn- EPs progress towards Ngn+ , then Fev+ and then branch towards the more committed lineages.
Interestingly we see two groups of Fev+ populations branching from the Ngn+ populations which
subsequently progress towards the distinct cell lines. This is consistent with an observation by
Bastidas-Ponce 2019 who also noted the emergence of two Fev+ cell types after the Ngn3+ state.

Fig. S28: VIA on 500HVG of Hematopoiesis

Fig. S28 Highly variable genes (HVGs) as unprocessed input using the scRNA-seq data of hematopoiesis (no
PCA) (a) VIA graph on 500 HVGs without PCA (b) unsupervised plotting of gene trends vs. inferred pseudotimes for
predicted lineages. (c) Coarse grained VIA graph. (d) VIA graph topology for increasing number of HVGs without
PCA show that the overall pronged topology is stable with erythroid and megakaryocytic branches on one side of the
HSCs, and the monocytes and DCs on the other side.

Fig. S29: VIA on HVG of Endocrine genesis

**Fig. S29 Highly variable genes (HVGs) as direct input using the scRNA-seq data of endocrine genesis (no**
 **PCA)** (a) Overall graph level topology of VIA, nodes colored by inferred pseudotime and automatically detected
 terminal nodes circled red. (b) Marker gene expression plotted for each lineage shows that each cell fate and
 pathway uniquely leads to the islet fate associated with that marker gene. (c) Topology of endocrine dataset for
 different number of HVGs, continues to represent the overall progression from EPs to bifurcation of *Fev*⁺ cells that
 lead to the different islets.

References

- 1. Chen, H., Albergante, L., Hsu, J.Y. et al. Single-cell trajectories reconstruction, exploration and
 mapping of omics data with STREAM. *Nat Commun* 10, 1903 (2019).
 <https://doi.org/10.1038/s41467-019-09670-4>
- 2. Saelens, W., Cannoodt, R., Todorov, H. et al. A comparison of single-cell trajectory inference
 methods. *Nat Biotechnol* 37, 547–554 (2019). <https://doi.org/10.1038/s41587-019-0071-9>
- 3. Gerard, D. Data-based RNA-seq simulations by binomial thinning. *BMC Bioinformatics* 21, 206
 (2020). <https://doi.org/10.1186/s12859-020-3450-9>
- 4. Stassen SV, Siu DMD, Lee KCM, Ho JWK, So HKH, Tsia KK. PARC: ultrafast and accurate
 clustering of phenotypic data of millions of single cells. *Bioinformatics*. 2020 May
 1;36(9):2778-2786. doi: 10.1093/bioinformatics/btaa042.
- 5. Qiu, P. Embracing the dropouts in single-cell RNA-seq analysis. *Nat Commun* 11, 1169 (2020).
 <https://doi.org/10.1038/s41467-020-14976-9>

- 6. Helena Todorov, Robrecht Cannoodt, Wouter Saelens, Yvan Saeys, TinGa: fast and flexible
trajectory inference with Growing Neural Gas, *Bioinformatics*, Volume 36, Issue Supplement_1,
July 2020, Pages i66–i74, <https://doi.org/10.1093/bioinformatics/btaa463>
- 7. Tran TN, Bader GD (2020) Tempora: Cell trajectory inference using time-series single-cell RNA
sequencing data. *PLoS Comput Biol* 16(9): e1008205.
<https://doi.org/10.1371/journal.pcbi.1008205>
- 8. Blasi, T., Hennig, H., Summers, H. et al. Label-free cell cycle analysis for high-throughput
imaging flow cytometry. *Nat Commun* 7, 10256 (2016). <https://doi.org/10.1038/ncomms10256>
- 9. Eulenberg P, Köhler N, Blasi T, Filby A, Carpenter AE, Rees P, Theis FJ, Wolf FA.
Reconstructing cell cycle and disease progression using deep learning. *Nat Commun*. 2017 Sep
6;8(1):463. doi: 10.1038/s41467-017-00623-3. PMID: 28878212; PMCID: PMC5587733.
- 10. Bastidas-Ponce, A. et al. Comprehensive single cell mRNA profiling reveals a detailed roadmap
for pancreatic endocrinegenesis. *Development* 146, (2019).

Reviewers' Comments:

Reviewer #1:

None

Reviewer #2:

Remarks to the Author:

The authors have systemically and methodically answered the questions I raised and appeared to have a similarly good job of replying to the other referees. I am happy to now recommend publication of this manuscript.

Reviewer #3:

Remarks to the Author:

Review of revised VIA manuscript:

R3.Q1: For the synthetic datasets, I am satisfied with the extra metrics that the authors included in the paper, and merged into their final composite metric, that in my opinion, allows a better comparison of the trajectories returned by the different tools.

For the real datasets, I value the addition of the correlation scores for common lineages. While I understand the authors choice not to rely on a fixed trajectory in real datasets (since we can never be completely sure of what the real topology is in those datasets), I nevertheless think that reusing the metrics that they defined for synthetic datasets might have been interesting on real datasets as well.

R3.Q2: The authors answered my concern

R3.Q3: The authors answered my question

R3.Q4: I now agree with the way the authors rephrased the paragraph

R3.Q5: I now agree with the way the authors describe and explain how a starting point can be defined for the VIA method.

R3.Q6 and Q7: The extra experiments that the authors conducted (especially on real datasets) and that are now presented in supplementary figures answer the remaining questions I had.

Manuscript ID: NCOMMS-21-04939B
Author responses to the Final Revisions

Reviewer #2 (Remarks to the Author):

The authors have systemically and methodically answered the questions I raised and appeared to have a similarly good job of replying to the other referees. I am happy to now recommend publication of this manuscript.

We are delighted to see that the Reviewer is satisfied with the revisions and would like to take this opportunity to thank them for their constructive and very thoughtful feedback which led to an improved manuscript.

Reviewer #3 (Remarks to the Author):

R3.Q1: For the synthetic datasets, I am satisfied with the extra metrics that the authors included in the paper, and merged into their final composite metric, that in my opinion, allows a better comparison of the trajectories returned by the different tools. For the real datasets, I value the addition of the correlation scores for common lineages. While I understand the authors' choice not to rely on a fixed trajectory in real datasets (since we can never be completely sure of what the real topology is in those datasets), I nevertheless think that reusing the metrics that they defined for synthetic datasets might have been interesting on real datasets as well.

We are grateful that the Reviewer appreciates our approach of using the composite metric for offering a more holistic comparison of different trajectories. We agree that the addition of metrics spanning topology and pseudotime have been a critical and necessary improvement to the validation of VIA and thank you for suggesting this. We note your recommendation to apply (perhaps with some modifications) this methodology towards assessing the topologies of real datasets. As part of our future work on VIA, we are considering extending the current composite accuracy framework so it can be applied to real data by formulating a reliable reference topology for the different datasets.

R3.Q2: The authors answered my concern

R3.Q3: The authors answered my question

R3.Q4: I now agree with the way the authors rephrased the paragraph

R3.Q5: I now agree with the way the authors describe and explain how a starting point can be defined for the VIA method.

R3.Q6 and Q7: The extra experiments that the authors conducted (especially on real datasets) and that are now presented in supplementary figures answer the remaining questions I had.

Questions 2-7: We would like to take this opportunity here to thank you for making time to evaluate all the revisions and providing suggestions that have strengthened the manuscript as well as the underlying VIA method.